# Impact of steroid differentiation on tumor microenvironment revealed by single-nucleus atlas of adrenal tumors

Anne Jouinot [1,2] ✉, Yoann Martin[1], Florian Violon[1,3], Thomas Foulonneau[1], Yanis Bendjelal[1], Philip Calvet [1], Brigitte Izac[1], Franck Letourneur [1], Céline Bertholle[1], Muriel Andrieu [1], Rachel Onifarasoaniaina[1], Maryline Favier [1], Charlène de Guitaut[4], Archibald Fraikin[4], Daniel de Murat[1], Roberta Armignacco[1], Nesrine Benanteur[1], Maria Francesca Birtolo [1], Mathilde Sibony [1,3], Karine Perlemoine[1], Patricia Vaduva [1], Lucas Bouys [1,2], Fidéline Bonnet-Serrano [1,5], Bertrand Dousset[1,6], Martin Gaillard[1,6], Eric Pasmant[1,7], Maxime Barat[1,8], Anthony Dohan[1,8], Magalie Haissaguerre[9], Antoine Tabarin[9], Rossella Libé[1,2], Laurence Guignat[2], Lionel Groussin[1,2], Annabel Berthon[1], Bruno Ragazzon [1], Jérôme Bertherat [1,2] & Guillaume Assié[1,2] ✉

Adrenocortical carcinomas (ACC) are aggressive and resistant to medical treatment. This study reports a single-nucleus transcriptome atlas of steroid and microenvironment cells in 38 human normal adrenals and adrenocortical tumors. We identify intermediate-state cells between glomerulosa and fasciculata, a transition state in the centripetal trans-differentiation of normal steroid cells. In tumors, steroid cells show expression programs reflecting this zonation. Although ACC microenvironment is scarce, its signatures combine with those of steroid cells into ecotypes. A first ecotype combines cancer-associated fibroblasts, tumor-associated endothelial cells, with hypoxia and mitosis signatures in steroid cells. Another ecotype combines exhausted T cells, with fasciculata steroid signature. These ecotypes are associated with poor survival. Conversely, a third ecotype combines inflammatory macrophages, with reticularis steroid signature, and better outcome. These steroid/microenvironment cells interplays improve outcome predictions and may open therapeutic options in aggressive ACC, through immune microenvironment activation by modulating glucocorticoids/androgens balance.

The adrenal cortex, corresponding to the external part of adrenal glands, is divided into three well differentiated anatomical and functional regions, which produce different steroid hormones. From periphery to center, the zona glomerulosa produces aldosterone, the zona fasciculata produces cortisol, and the zona reticularis produces androgens[1].

Primary tumors arising from the adrenal cortex include common benign unilateral tumors -referred to as adrenocortical adenomas

(ACA)-, rare malignant tumors -referred to as adrenocortical carcinomas (ACC)-, and multiple nodules on both adrenals -referred to as primary bilateral macronodular adrenocortical hyperplasia (PBMAH)-. Morbidity of adrenal tumors is high, related either to steroid hormone excess in benign and malignant tumors, or to tumor growth and metastases in ACC[2,3].

Bulk genomic studies have identified distinct molecular classes for adrenocortical tumors. ACC cluster in two distinct transcriptome classes, named C1A with poor outcome and C1B with better outcome[4,5]. Transcriptome signatures show an enrichment of steroidogenic and cell-cycle related genes in C1A, and of immune related genes in C1B[6,7]. Benign tumors cluster in several distinct groups, following their somatic or germline genetic driver mutations[8,9].

Adrenocortical tumors present a limited proportion of stromal and immune cells[10], with poorly established prognostic and pathophysiologic relevance so far. Single-cell tumor atlases recently emerged as a powerful tool for characterizing tumor microenvironment and intra-tumor heterogeneity[11–13]. However, single-cell studies on the human adrenal cortex are limited, focusing on normal adrenal cortex[14], and on the steroid cells of adrenocortical tumors[15–17].

In this work, we provide an atlas of human normal and tumoral adrenals, covering normal and tumor steroid cells, and microenvironment cells in a series of different tumor types. We show original steroid and microenvironment cell signatures, integrated into ecotypes associated with outcome in ACC.

## Results

### Single-nucleus atlas of human adrenal cortex reveals cancer-specific microenvironment signatures

To elucidate the cellular architecture of adrenal tumors, we analyzed 4 normal adrenals and 34 primary tumors, including 11 C1A ACC, 9 C1B ACC, 8 adenomas and 6 PBMAH (Supplementary Data 1-2). A total of 168,927 cells passed quality control (Supplementary Fig. 1a–e). Cells were annotated using canonical lineage markers and cell type predictions based on published gene signatures[11,18] (Fig. 1a, b, Supplementary Data 3, Supplementary Fig. 1f, g). All major cell types were represented across all samples (Fig. 1c). Uniform manifold approximation and projection (UMAP) visualization showed a clear separation of steroid cells by tumor, with a tendency of tumors to aggregate according to the normal/benign/malignant status, and according to the tumor types previously reported[6–8] (Fig. 1d, e). In contrast, UMAP visualization of stromal and immune cells across tumors clustered together without batch correction (Fig. 1a, d, e). Remarkably, stromal and myeloid cells were distributed along gradients reflecting malignancy (Fig. 1a, d, e).

We estimated single-nucleus copy number variant (CNV) profiles using inferCNV[19]. As expected, neoplastic cells cumulated large-scale genomic rearrangements (Fig. 1f, Supplementary Fig. 1h), including chromosome 5, 7, 12 gains and 1, 18, and 22 losses commonly reported in ACC[6,7], or the recurrent 1p losses in *KDM1A*-mutated PBMAH[9,20].

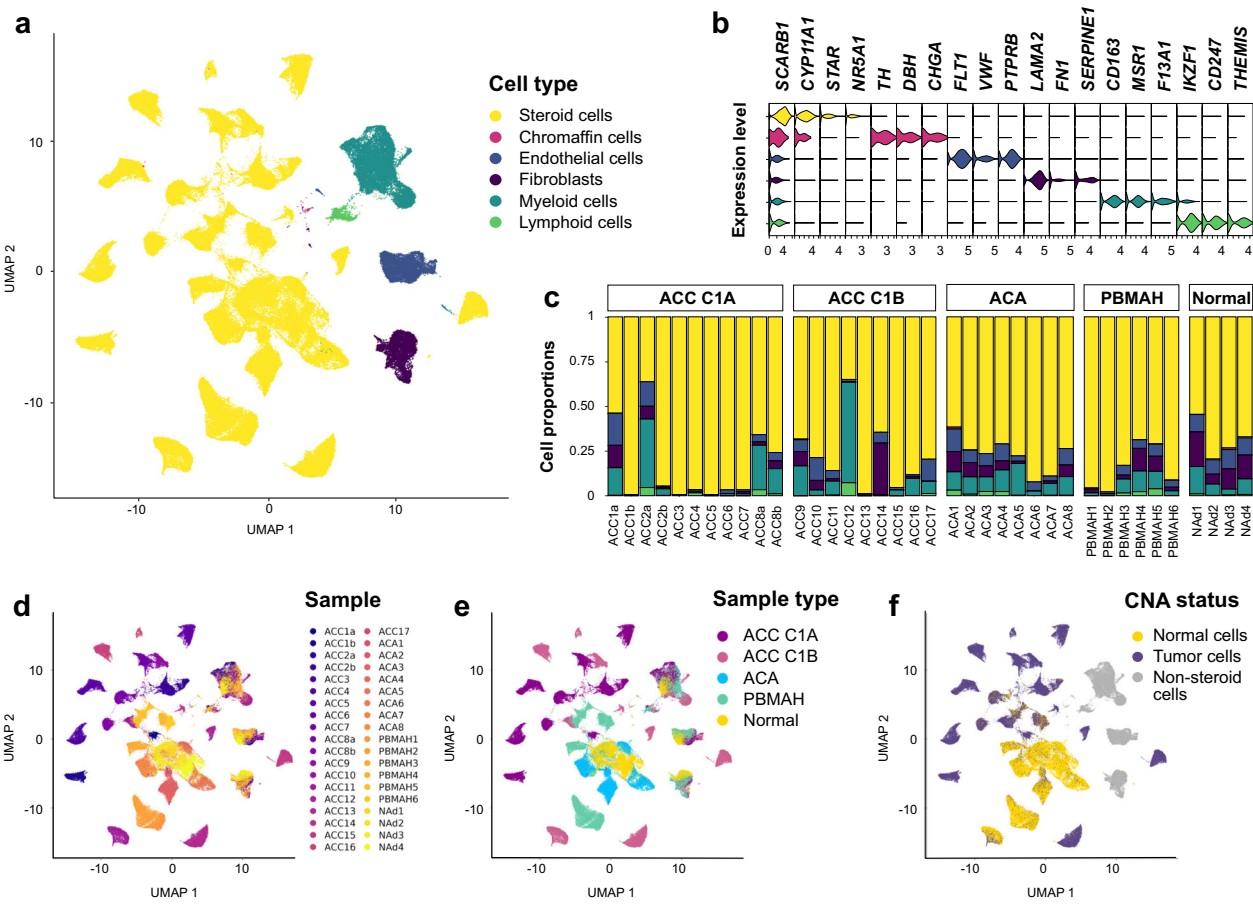

**Fig. 1 | Cell atlas of normal adrenal cortex and adrenocortical tumors.** Individual cells from 34 adrenocortical tumors and 4 normal adrenals are presented (single-nucleus transcriptomes). **a** Uniform manifold approximation and projection (UMAP) annotated with cell types. **b** Stacked violin plot of cell types markers, showing log-scaled raw counts for top differentially expressed genes for each cell type. **c** Proportion of cell types in each sample, presented by sample type: adrenocortical carcinoma (ACC), adrenocortical adenoma (ACA), primary bilateral macronodular adrenocortical hyperplasia (PBMAH) or normal adrenal. **d** UMAP annotated by sample identifier. **e** UMAP annotated by sample type. **f** UMAP annotated by benign/malignant statuses, based on copy number alterations (CNA) scores inferred from transcriptome. Tumor cells were called if >3% of the genome was altered.

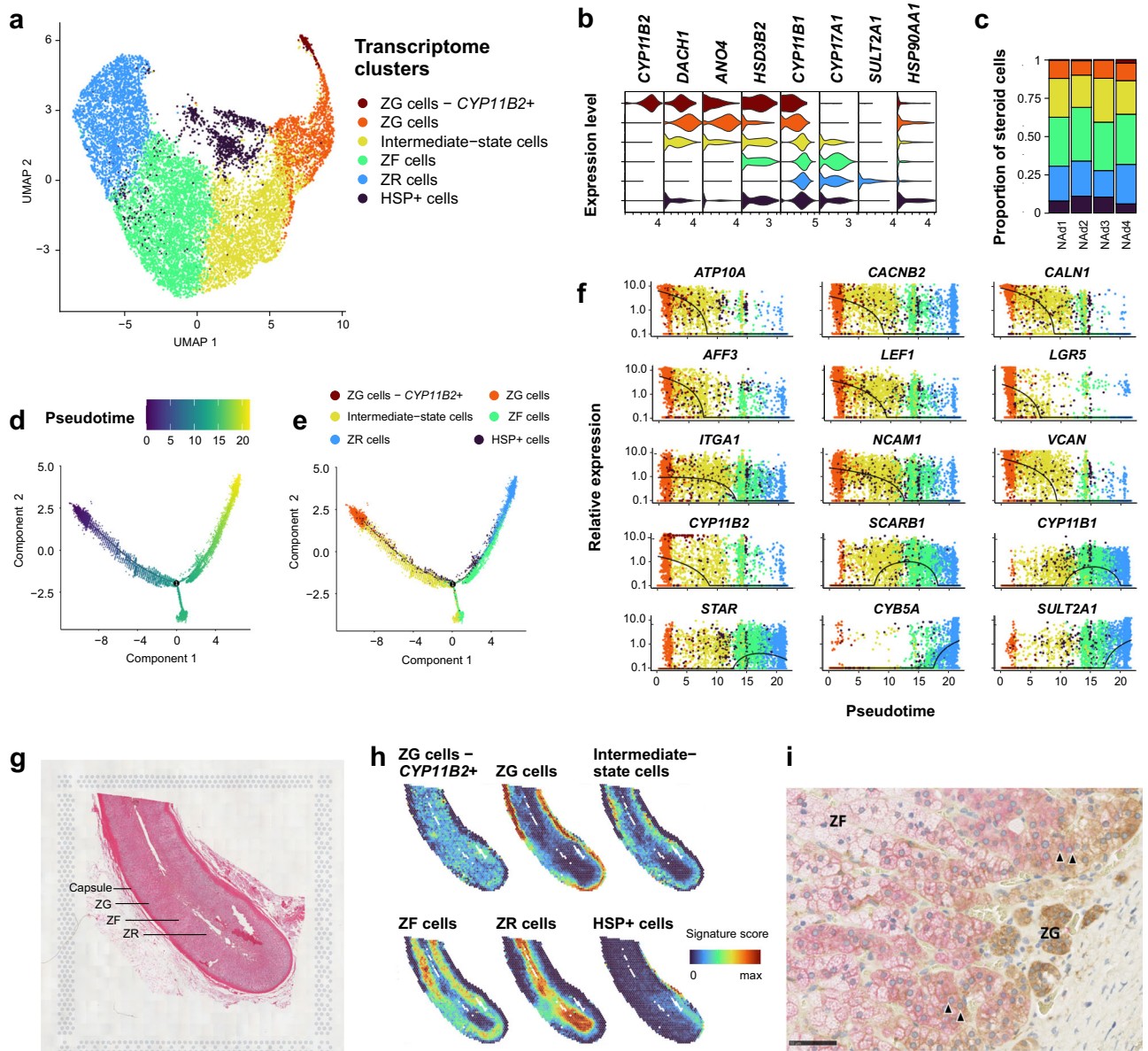

**Fig. 2 | Characterization of normal steroid cells.** Individual steroid cells from 4 normal adrenals are presented (single-nucleus transcriptomes). **a** UMAP annotated by transcriptome clusters; clustering and projection were performed after integration of samples (Seurat PrepSCTIntegration, FindIntegrationAnchors, and IntegrateData functions). **b** Stacked violin plot of transcriptome cluster markers, showing log-scaled raw counts for top differentially expressed genes for each cluster. **c** Proportion of transcriptome clusters in each sample. **d** Steroid cells trajectory annotated by pseudotime, using discriminative dimensionality reduction via learning a tree (DDRtree). **e** Steroid cells trajectory annotated by transcriptome clusters, using DDRtree. **f** Expression profiles of selected genes with pseudotime variation, including genes related to calcium signaling (*ATP10A, CACNB2, CALN1*), to

Wnt-βcatenin signaling (*AFF3, LEF1, LGR5*), to cell adhesion (*ITGA1, NCAM1, VCAN*), and to steroidogenesis (*CYP11B2, SCARB1, CYP11B1, STAR, CYB5A, SULT2A1*). Black lines: mean expression over pseudotime. **g** Hematoxylin eosin coloration of NAd4, a normal adrenal sample used for spatial transcriptomics. Capture area 6.5 × 6.5 mm. **h** Spatial transcriptomic representation of the 6 signatures of normal steroid cells in NAd4. Normal steroid cells signatures were deconvoluted with Cell2location in spatial transcriptomics spots. **i** DAB2 (ZG marker, brown) and CYP17A1 (ZF/ZR marker, red) immunohistochemistry staining of a normal adrenal (magnification x400, barscale 50 μm). Some cells co-expressed DAB2 and CYP17A1 (arrowheads). Abbreviations: ZG zona glomerulosa, ZF zona fasciculata, ZR zona reticularis, HSP heat-shock proteins.

## Normal human adrenal cortex differentiates following the adrenal zonation

Normal steroid cells gathered into zona glomerulosa, zona fasciculata, and zona reticularis clusters, in agreement with the adrenal cortex functional zonation (Fig. 2a–c, Supplementary Fig. 2a–f). An intermediate-state cluster was identified, showing features of both zona glomerulosa and zona fasciculata (Fig. 2a–c, Supplementary Fig. 2d, Supplementary Data 4), as well as an enriched expression of cell adhesion coding genes (*ITGA1, NCAM1*) (Supplementary Data 4, Supplementary Fig. 2e). A last cluster was individualized, characterized

by high expression of heat shock proteins (HSP, Supplementary Data 4, Supplementary Fig. 2e).

In order to estimate the gradual transition of adrenocortical cells, cells were sorted according to pseudotime, revealing a trajectory starting from zona glomerulosa cells to zona fasciculata and zona reticularis cells. In this trajectory, the intermediate-state cluster appeared between zona glomerulosa and zona fasciculata cells, and the HSP+ cluster was disseminated within intermediate-state and zona fasciculata clusters (Fig. 2d, e). Genes associated with pseudotime reflected the specific gene expression pattern observed in the adrenal

cortex functional zonation, starting with high expression of zona glomerulosa markers (*CYP11B2*, calcium signaling, and Wnt-βcatenin signaling genes), followed by zona fasciculata markers (*CYP11B1* and other steroidogenesis related genes) and then zona reticularis markers[21] (*CYB5A, SULT2A1*) (Fig. 2f, Supplementary Data 5).

In order to precise the in-situ localization of steroid cell types, we performed spatial transcriptomics in two normal adrenals. The different adrenal cell types could be identified by unsupervised clustering (Supplementary Fig. 2g–i, Supplementary Data 6). Deconvolution of single-nucleus clusters properly localized zona glomerulosa, zona fasciculata, and zona reticularis cells. In accordance with the pseudotime analysis, the intermediate-state cells were located between zona glomerulosa and zona fasciculata (Fig. 2g, h, Supplementary Fig. 2j, k). Of note, the proportion of intermediate-state cells in spatial transcriptomics and sn-RNAseq was variable, depending on clustering parameters. This variability may reflect the unclear delineation of intermediate-state cells in a progressive trans-differentiation from ZG to ZF. These intermediate-state cells could be independently validated by immunohistochemistry staining, showing co-expression of ZG marker DAB2 and ZF/ZR marker CYP17A1 (Fig. 2i). HSP+ cells were concentrated into a focused area, within zona fasciculata (Fig. 2g, h, Supplementary Fig. 2j, k).

### Recurrent gene modules drive steroid cell heterogeneity

In steroid tumor cells, 8 recurrent gene modules drove neoplastic cell heterogeneity (Fig. 3a, Supplementary Data 7). Four gene modules corresponded to steroid cells signatures found in normal adrenals, one enriched in markers of ZG cells (GM1_ZG), two enriched in markers of ZF cells (GM2_ZF1 and GM3_ZF2) and one enriched in markers of ZR cells (GM4_ZR, Fig. 3b). The four remaining gene modules were related to tumor signatures, enriched in extracellular matrix genes (GM5_ECM), translation-related genes (GM6_Translation), mitosis-related genes (GM7_Mitosis), and hypoxia-related genes (GM8_Hypoxia) respectively (Fig. 3b).

Each of the 8 gene modules was scored in each cell of each adrenal tumor sample. Each cell was labeled following the maximal gene module score. Cancer cells were enriched in GM7_Mitosis and GM8_Hypoxia, while benign tumor cells were enriched in GM5_ECM (Fisher simulated $p = 0.0005$, Fig. 3c, d). The proportion tumor steroid cells expressing GM7_Mitosis module correlated with tumor mitotic count (Spearman $r = 0.59$, $p = 0.003$) and Ki-67 index (Spearman $r = 0.58$, $p = 0.005$, Supplementary Fig. 3a, b). The proportion of tumor steroid cells expressing GM3_ZF2 module correlated with cortisol secretion in patients (Spearman $r = 0.63$, $p = 0.01$, Supplementary Fig. 3c). These correlations confirmed the biological relevance of gene modules.

To evaluate the clinical relevance of these signatures, each gene module was scored in bulk transcriptomes from three ACC datasets[6,7,22]. GM7_Mitosis was positively correlated with GM8_Hypoxia (Pearson $r = 0.38$, $p < 10^{-7}$) and negatively correlated with GM4_ZR (Pearson $r = -0.44$, $p < 10^{-10}$, Supplementary Fig. 3d). Gene modules were associated with several clinical parameters, including GM3_ZF2 with cortisol secretion in patients (Spearman $r = 0.43$, $p < 10^{-8}$), and GM7_Mitosis with tumor mitotic count and Ki-67 index (Spearman $r = 0.48$ and 0.36, and $p < 10^{-8}$ and $<10^{-4}$ respectively, Supplementary Fig. 3d). Six of the 8 gene modules were associated with disease-free and overall survival (Cox $p < 0.001$), with a better outcome for GM4_ZR, and a worse outcome for GM1_ZG, GM2_ZF1, GM3_ZF2, GM6_Translation and GM7_Mitosis (Fig. 3e, Supplementary Data 8).

### Cancer-associated stromal cells combine pan-cancer and adrenal-specific features

Fibroblasts and endothelial cells represented 4.7% and 5.1% of total cells, respectively (Fig. 1a).

Unsupervised clustering segregated resident fibroblasts -predominant in normal adrenals-, and cancer-associated fibroblasts (CAF) (Fig. 4a–c, Supplementary Fig. 4a). CAF were characterized by enrichment in extracellular matrix (ECM) remodeling and cell-matrix adhesion processes (Supplementary Fig. 4b, Supplementary Data 9). CAF were further clustered in three groups (CAF1, CAF2, CAF3), each with specific marker genes (Fig. 4b).

Sorting cells according to pseudotime revealed a branched trajectory starting from resident fibroblasts, towards CAF1 or towards CAF2 and CAF3 (Fig. 4d, e). Genes associated with pseudotime included ECM remodeling markers[23] towards CAF1 (*VCAN, BNC2*, Fig. 4f), and pro-angiogenic[24] (*RGS5, SEMA5A*) and immunosuppressive markers[25] towards CAF3 (*CD36*, Fig. 4g, Supplementary Data 10).

Endothelial cells clustered into the three canonical endothelial types -lymphatic, arterial, and venous- (Fig. 5a–c, Supplementary Fig. 5a). A subset of venous endothelial cells expressed high levels of heat shock proteins (EC-HSP+). In addition, two clusters were almost exclusively observed in adrenocortical tumors (Fig. 5a–c). One cluster -tumor endothelial cells 1 (TEC1)- included both ACC and benign adrenocortical tumors, whereas the other -tumor endothelial cells 2 (TEC2)- was restricted to ACC (Fisher simulated $p = 0.0005$). These two clusters were characterized by high expression of pro-angiogenic markers previously reported in tumor endothelial cells, such as *NRP1* in TEC1[26], and *VWF* and *ANGPT2* in TEC2[27,28] (Fig. 5b, Supplementary Fig. 5b, Supplementary Data 11).

Sorting cells according to pseudotime revealed a branched trajectory starting from venous endothelial cells, towards lymphatic endothelial cells or towards TEC2 (Fig. 5d, e). Genes associated with pseudotime included known TEC-associated genes (*VWF, ANGPT2*) towards TEC2, as well as markers not described before (*ANO2, KCNQ3, LAMB1, ENPP2*, Fig. 5f, Supplementary Data 12).

TEC2 signature was deconvoluted in bulk transcriptome from three ACC datasets, showing an association with poor outcome (Fig. 5g, Supplementary Data 8).

### Better-outcome adrenocortical carcinomas are enriched in inflammatory macrophages

Lymphoid and myeloid cells represented 1.2% and 9.9% of total cells, respectively (Fig. 1c).

Lymphoid cells were mainly distributed into four T cells clusters, including naive/memory -expressing *IL7R*-, exhausted -expressing higher levels of *TOX*- and NK-like[29] -expressing *GNLY* and *KLRF1*-, and a group of unassigned T cells (Fig. 6a–c, Supplementary Fig. 6a, b, Supplementary Data 13). Exhausted and NK-like T cells were predominant in ACC, while unassigned T cells were predominant in normal adrenal and benign tumors (Fisher simulated $p = 0.0005$, Fig. 6c).

Sorting cells according to pseudotime revealed a trajectory starting from unassigned towards NK-like or exhausted T cells (Fig. 6d, e). Genes associated with pseudotime included NK differentiation markers towards NK-like T cells (*KLRD1, KLRF1* and *GNLY*, Fig. 6f), and *THEMIS*, a negative regulator of effector CD8+ T cells[30], towards exhausted T cells (Fig. 6g, Supplementary Data 14).

Myeloid cells clustered into resident -predominant in normal adrenals-, inflammatory - expressing *C3, CX3CR1*, and MHCII markers-, perivascular[31] -expressing *SELENOP* and *LYVE1*- and cycling macrophages[32] -expressing *MKI67* and *TOP2A*-, and two groups of tumor-associated macrophages (TAM1 and TAM2, Fig. 7a–c, Supplementary Fig. 7a). TAM markers were validated in situ using high definition spatial transcriptomics and immunohistochemistry staining (Fig. 7d, e, Supplementary Fig. 8a, b). Resident macrophages and TAM expressed high levels of "M2" polarization markers[33] (*CD163* and *MRC1*), and of *FKBP5*, reflecting glucocorticoid receptor activation[34,35]. Conversely, inflammatory macrophages showed low expression of *CD163, MRC1* and *FKBP5* (Fig. 7b, Supplementary Fig. 7b, Supplementary Data 15).

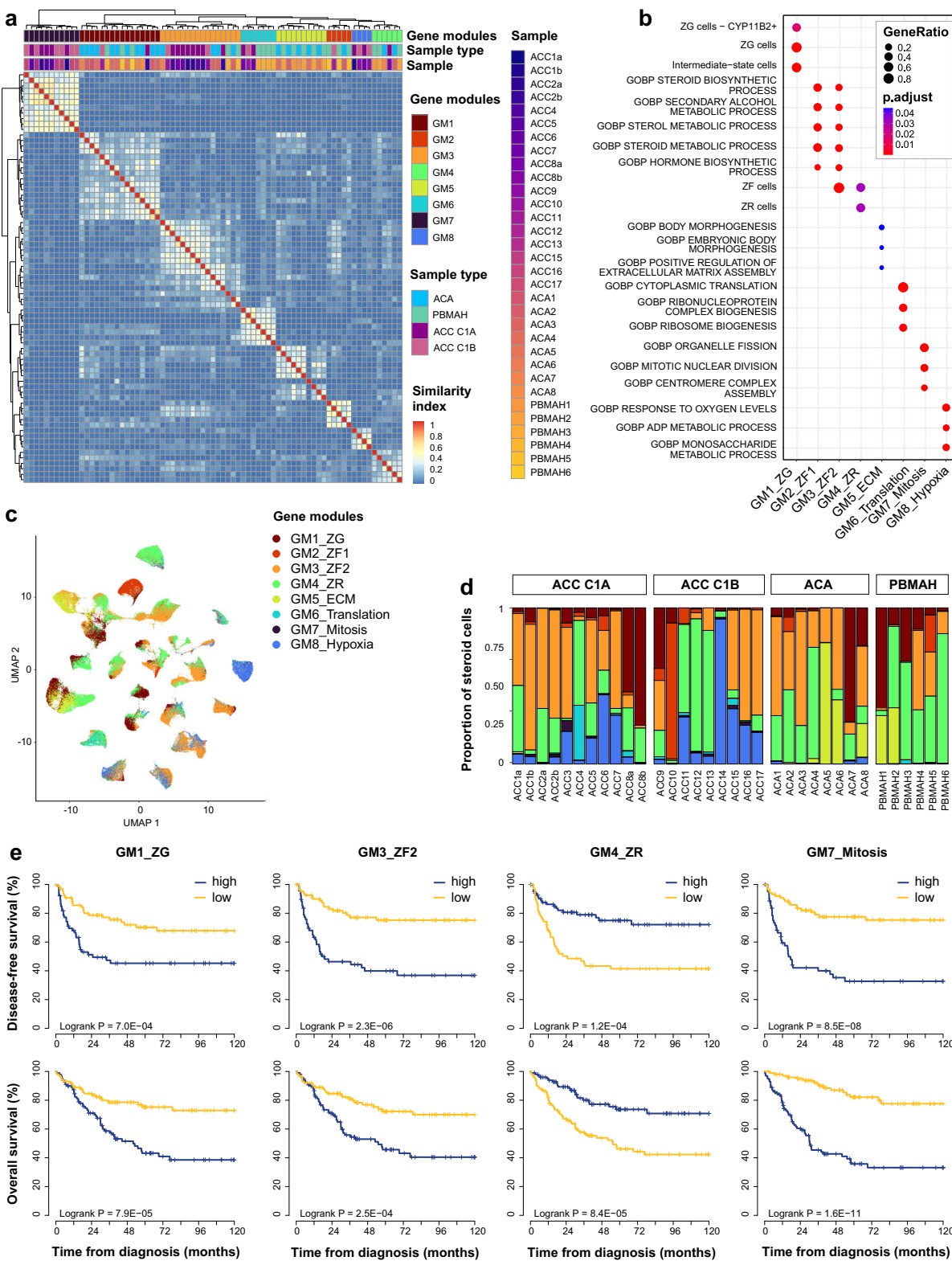

**Fig. 3 | Characterization of tumor steroid cells. a** Similarity heatmap of gene modules. For each tumor, gene expression variability was captured by the 10 first PCA components. From each PCA component, gene modules were generated (top 50 genes positively and negatively associated with each component). The heatmap of gene modules similarities (Sørensen indexes) revealed 8 recurrent gene modules. **b** Gene set enrichment (normal adrenal signatures and GO-BP, over-representation test) of the 8 recurrent gene modules. **c** UMAP of the steroid tumor cells. Individual steroid cells from 34 adrenocortical tumors are presented (single-nucleus transcriptomes). The main gene module is assigned to each cell. **d** Proportion of main gene modules assigned to steroid cells of each tumor sample. **e** Association of gene modules and outcome. Gene modules were scored in bulk ACC transcriptomes from 201 patients with ssGSEA. Kaplan–Meier curves represent disease-free and overall survival.

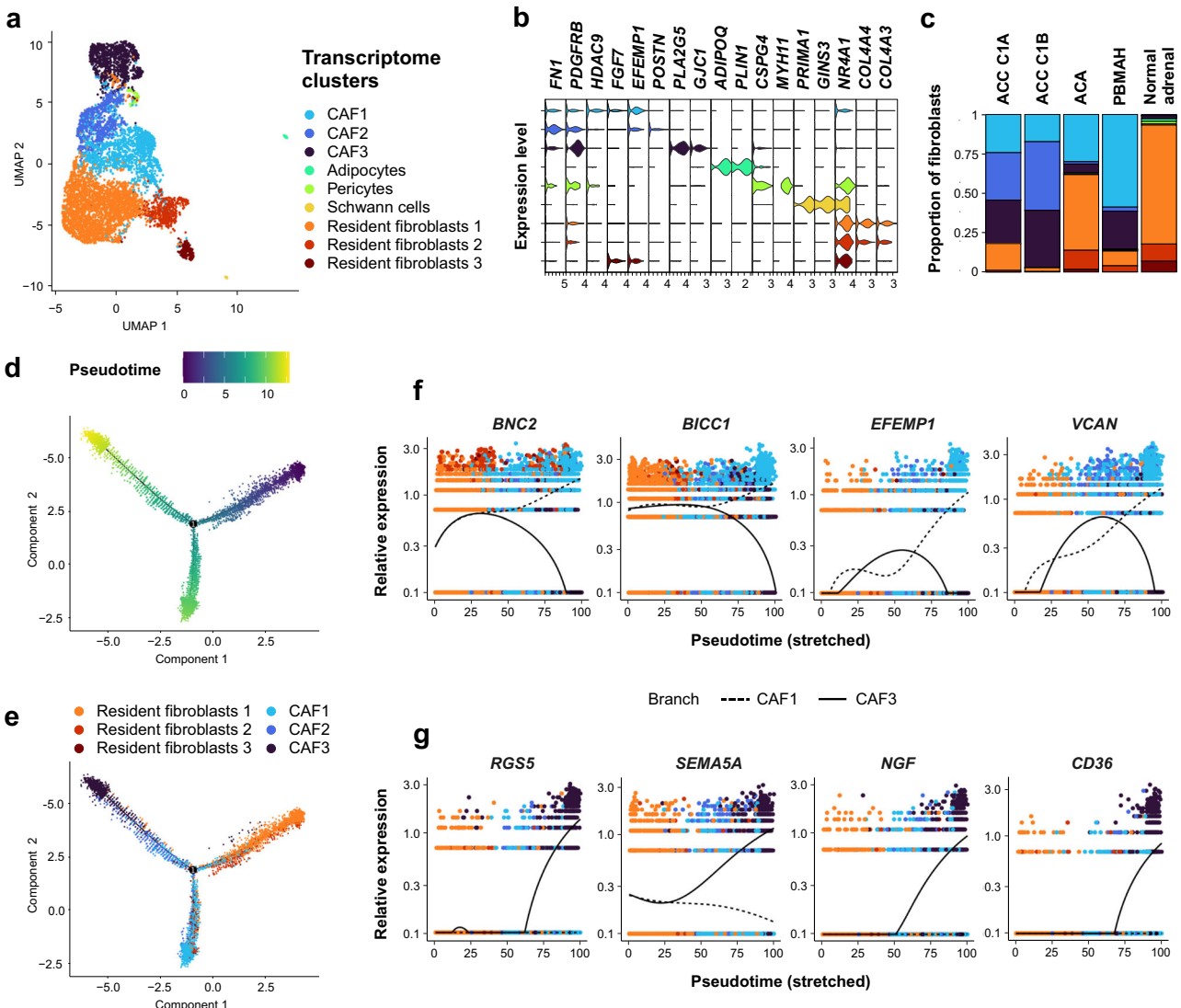

**Fig. 4 | Characterization of adrenocortical fibroblasts.** Individual fibroblasts cells from 34 adrenocortical tumors and 4 normal adrenals are presented (single-nucleus transcriptomes). **a** UMAP annotated by transcriptome clusters. **b** Stacked violin plot of transcriptome cluster markers, showing log-scaled raw counts for top differentially expressed genes for each cluster. **c** Proportion of transcriptome clusters in each sample type. **d** Fibroblasts trajectory annotated by pseudotime, using discriminative dimensionality reduction via learning a tree (DDRtree).

**e** Fibroblasts trajectory annotated by transcriptome clusters, using DDRtree. **f** Expression profiles of top genes with pseudotime variation from resident fibroblasts towards CAF1 (dashed line), including ECM remodeling markers (*VCAN, BNC2*). **g** Expression profiles of top genes with pseudotime variation from resident fibroblasts towards CAF3 (solid line), including pro-angiogenic (*RGS5, SEMA5A*) and immunosuppressive markers (*CD36*). Black lines: mean expression over pseudotime.

Sorting cells according to pseudotime revealed a branched trajectory starting from resident macrophages towards either TAM1 or inflammatory macrophages (Fig. 7f, g). Genes associated with pseudotime included M2/TAM markers[33,36,37] (*PPARG, GPNMB, CCL18*) towards TAM1, as well as anti-inflammatory markers such as cholesterol transporters[38] (*ABCA1, ABCG1*) and the metalloproteinase *MMP19*, degrading the cytokine CX3CL1[39] (ligand of CX3CR1, Fig. 7h). Genes associated with pseudotime included pro-inflammatory markers (*CX3CR1, C3, PRKG1*) towards inflammatory macrophages, as well as the adhesion G protein-coupled receptor *ADGRB3* (Fig. 7i, Supplementary Data 16).

Using deconvolution in bulk transcriptome, inflammatory macrophages signature was negatively correlated with GM3_ZF2 score - reflecting cortisol secretion (Supplementary Fig. 3c, e) - (Pearson $r = -0.43$, $p < 10^{-9}$, Supplementary Fig. 7c), probably reflecting the immunosuppressive effect of glucocorticoids. Inflammatory macrophages signature was associated with better outcome in three

independent ACC datasets (Fig. 7j). The prognostic value of inflammatory macrophages was further tested against other microenvironment signatures and against established prognostic factors (Supplementary Data 8). In stepwise multivariable models (for DFS and for OS), the prognostic value of inflammatory macrophages remained, while other microenvironment signatures were discarded. In these models, inflammatory macrophages, GM7_Mitosis score - reflecting tumor grade (Supplementary Fig. 3a, b) -, and ENSAT tumor stage were independent prognostic factors of DFS and OS (Supplementary Data 8).

**Tumor regions with distinct aggressive features correspond to intra-tumor variations of steroid and microenvironment signatures**

Two features of tumor heterogeneity were explored: (i) the metastatic spreading, by comparing primary tumor (ACC1a) and a metachronous metastasis (ACC1b) in one patient (Supplementary Fig. 9a);

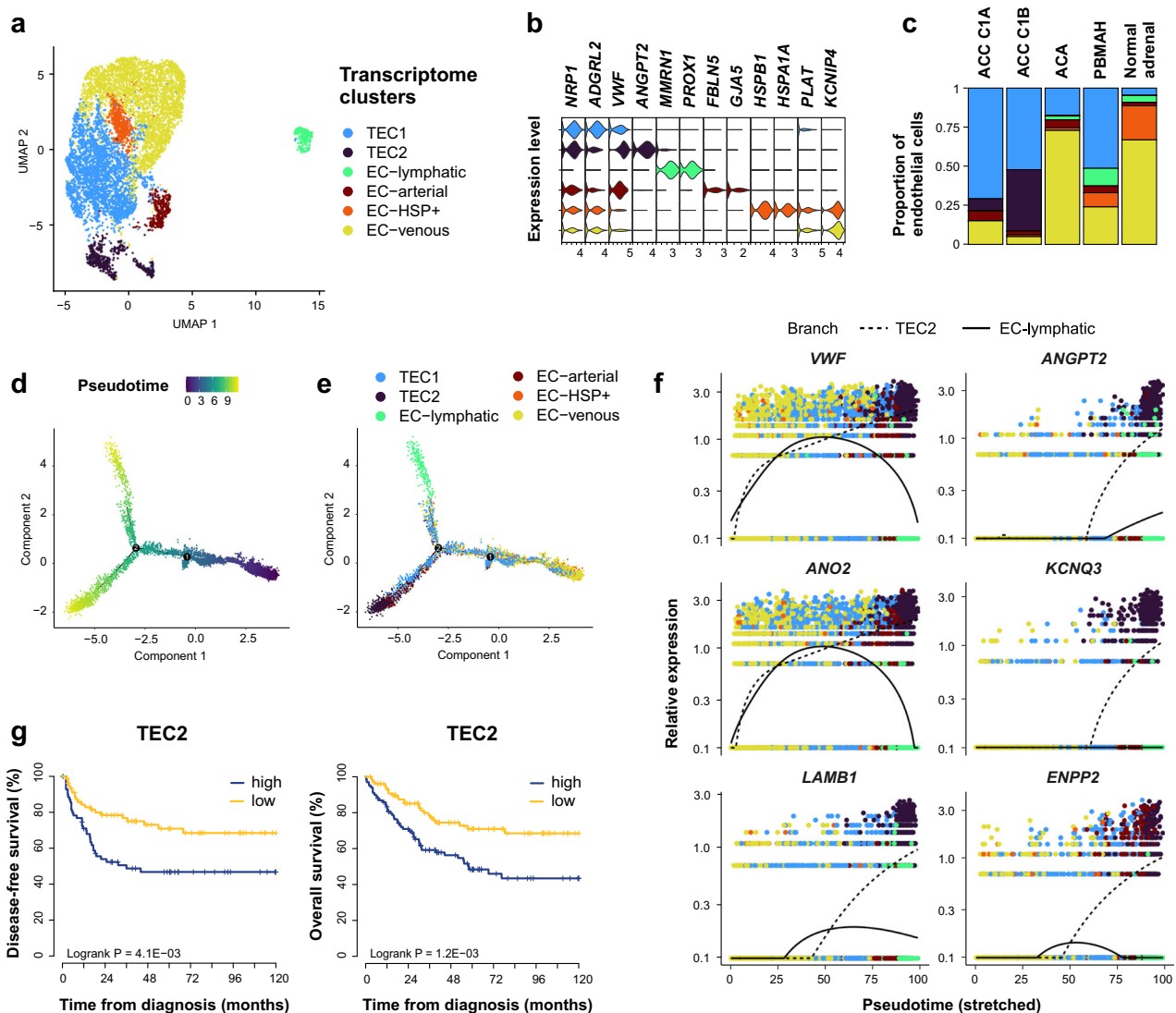

**Fig. 5 | Characterization of adrenocortical endothelial cells.** Individual endothelial cells from 34 adrenocortical tumors and 4 normal adrenals are presented (single-nucleus transcriptomes). **a** UMAP annotated by transcriptome clusters. **b** Stacked violin plot of transcriptome cluster markers, showing log-scaled raw counts for top differentially expressed genes for each cluster. **c** Proportion of transcriptome clusters in each sample type. **d** Endothelial cells trajectory annotated by pseudotime, using discriminative dimensionality reduction via learning a tree (DDRtree). **e** Endothelial cells trajectory annotated by transcriptome clusters, using DDRtree. **f** Expression profiles of top genes with pseudotime variation from EC-venous towards TEC2 (dashed line), including known TEC-associated genes (*VWF*, *ANGPT2*) and novel markers (*ANO2, KCNQ3, LAMB1, ENPP2*). Black lines: mean expression over pseudotime. **g** Association of TEC2 signature and outcome. TEC2 signature was deconvoluted in bulk ACC transcriptomes from 201 patients with CIBERSORTx. Kaplan–Meier curves represent disease-free and overall survival.

(ii) the tissue morphology, by comparing one tumor region with aggressive features (ACC2b and ACC8a) and another without (ACC2a and ACC8b) in two different patients (Supplementary Fig. 9b). Tumor heterogeneity was associated with variations in the microenvironment (Fig. 1c) and gene module composition (Fig. 3d, Supplementary Data 17). In ACC1, the metastasis sample (ACC1b) showed a lower proportion of microenvironment cells (<1% of cells, Fisher simulated $p = 0.0005$) and an enrichment in gene modules reflecting aggressive steroid cells (higher GM3_ZF2 and GM7_mitosis and lower GM4_ZR, Fisher simulated $p = 0.0005$, Supplementary Data 17). In ACC2, the region with more aggressive morphological features (ACC2b) showed a lower proportion of microenvironment cells (<10% of cells, Fisher simulated $p = 0.0005$) and an enrichment in TEC2 and GM3_ZF2 (Fisher simulated $p = 0.0005$, Supplementary Data 17). In ACC8, variations related to aggressive features in the microenvironment and in gene modules were more limited (Supplementary Data 17).

ACC2 was further characterized using spatial transcriptomics. Unsupervised clustering of spatial spots revealed three compartments (cluster1, cluster2, cluster3), corresponding to spatially distinct tumor regions, with different levels of cyto-nuclear atypia (Supplementary Fig. 9c–e). High level of chromosome alterations confirmed the malignant nature of cells (Supplementary Fig 9f). Cluster3 was enriched in immune cells while cluster1 and cluster2 were enriched in steroid cells (Supplementary Fig. 9g, h, Supplementary Data 18). Similarly, deconvolution of single-nucleus signatures showed an enrichment in TAM1, inflammatory macrophages, and NK-like T cells in cluster3 spots. In cluster1 and cluster2, spots were enriched in steroidogenic modules GM2_ZF1 and GM3_ZF2 (Supplementary Fig. 9i). Cluster1 and cluster2 differed by their enrichment in CAF1 and inflammatory macrophages, higher in cluster2 (Supplementary Fig. 9i).

Altogether, these results support that regions with aggressive features present higher levels of module GM3_ZF2 and GM7_mitosis with rather low immune cells.

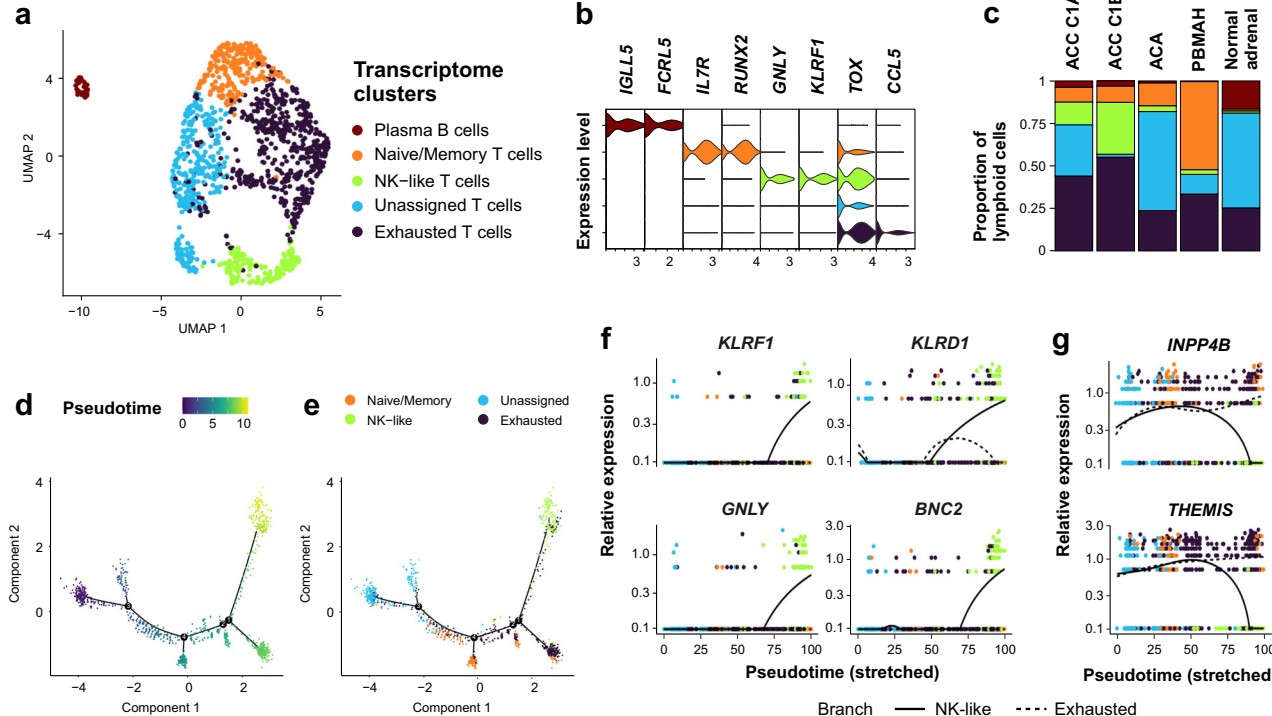

**Fig. 6 | Characterization of adrenocortical lymphocytes.** Individual lymphoid cells from 34 adrenocortical tumors and 4 normal adrenals are presented (single-nucleus transcriptomes). **a** UMAP annotated by transcriptome clusters. **b** Stacked violin plot of transcriptome cluster markers, showing log-scaled raw counts for top differentially expressed genes for each cluster. **c** Proportion of transcriptome clusters in each sample type. **d** Lymphocytes trajectory annotated by pseudotime, using discriminative dimensionality reduction via learning a tree (DDRtree).

**e** Lymphocytes trajectory annotated by transcriptome clusters, using DDRtree. **f** Expression profiles of top genes with pseudotime variation, from unassigned T cells towards NK-like T cells (solid line), including NK differentiation markers (*KLRD1*, *KLRF1* and *GNLY*). Black lines: mean expression over pseudotime. **g** Expression profiles of top genes with pseudotime variation, from unassigned T cells towards exhausted T cells (dashed line), including *THEMIS*, a negative regulator of TCR signaling. Black lines: mean expression over pseudotime.

## Single-nucleus signatures combine into ecotypes associated with outcome

Using deconvolution in bulk transcriptomes, the main single-nucleus signatures were scored in 201 ACC. Five clusters of signatures, referred to as ecotypes, were identified, including Eco1 reflecting mitosis and hypoxia with TEC2, Eco2 reflecting glucocorticoid steroidogenesis differentiation with exhausted T cells, Eco5 reflecting androgen steroidogenesis differentiation with inflammatory macrophages, Eco3 and Eco4 combining miscellaneous immune and endothelial cells (Fig. 8a, b). Of note, Eco1 mitosis/hypoxia and Eco2 ZF-like were associated with tumor grade (t-test statistic −4.63, $p < 10^{-4}$) and cortisol secretion (t-test statistic −5.90, $p < 10^{-7}$) respectively, supporting the biological relevance of these signatures. Beyond ecotypes, this clustering of single-nucleus signatures also discriminated two groups of tumors, with significant enrichment in C1A and C1B ACC respectively (Fisher $p < 10^{-15}$). C1A-like group was enriched in Eco1 mitosis/hypoxia and Eco2 ZF-like (t-test statistic 9.79, $p < 10^{-15}$ and statistic 11.69, $p < 10^{-15}$ respectively), while C1B-like group was enriched in Eco3 and Eco4 miscellaneous (t-test statistic −2.67, $p = 0.008$ and statistic −4.36, $p < 10^{-4}$ respectively), and in Eco5 ZR-like (t-test statistic −8.76, $p < 10^{-14}$).

Ecotypes were further explored in terms of cell-cell interactions using CellChat[40] in single-nucleus transcriptomes. In Eco1, interactions of GM8_hypoxia steroid cells with TEC2 or with CAF3 were enriched in ligand-receptor pairs related to angiogenesis pathways (Supplementary Fig. 10a, b, Supplementary Data 19). In Eco2, cell-cell interactions were enriched in ligand-receptor pairs related to cell adhesion pathways (Supplementary Fig. 10a, c, Supplementary Data 19). In Eco5, interactions of GM4_ZR steroid cells with inflammatory macrophages

or with CAF1 were enriched in ligand-receptor pairs related to inflammation pathways and immune activation (Supplementary Fig. 10a, Fig. 8c, Supplementary Data 19).

The prognostic value of ecotypes was explored in bulk transcriptomes. Eco1 mitosis/hypoxia and Eco2 ZF-like were associated with poorer outcome, while Eco5 ZR-like was associated with better outcome (Fig. 8d, Supplementary Data 20). The prognostic value of ecotypes was further tested in stepwise multivariable models including ENSAT stage. Eco1, Eco2, and stage were independent prognostic factors of DFS and OS (Supplementary Data 20). Finally, compared to models based only on clinical variables, models including ecotypes were better predictors for both DFS (C-index 0.808 vs 0.728, likelihood ratio test LRT $p < 10^{-4}$) and OS (C-index 0.841 vs 0.820, LRT $p < 10^{-3}$, Supplementary Data 21).

## Discussion

This study yields significant insights into the normal adrenal cortex and its pathophysiology through the single nucleus characterization of benign and malignant adrenocortical tumors.

In normal steroid cells, this study reveals an intermediate state signature that falls in between zona glomerulosa and zona fasciculata both functionally -as shown by the pseudotime trajectory-, and anatomically -as shown by spatial transcriptomics. Pseudotime trajectory analysis also shows a continuation between zona fasciculata and zona reticularis. Taken together these data are supporting in humans the kinetic lineage model of adrenocortical cells starting from the capsule towards the adrenal centers, as established in mouse models[41,42]. Normal adrenal analysis also identifies an original cluster of HSP+ steroid cells, projecting within the intermediate and zona fasciculata

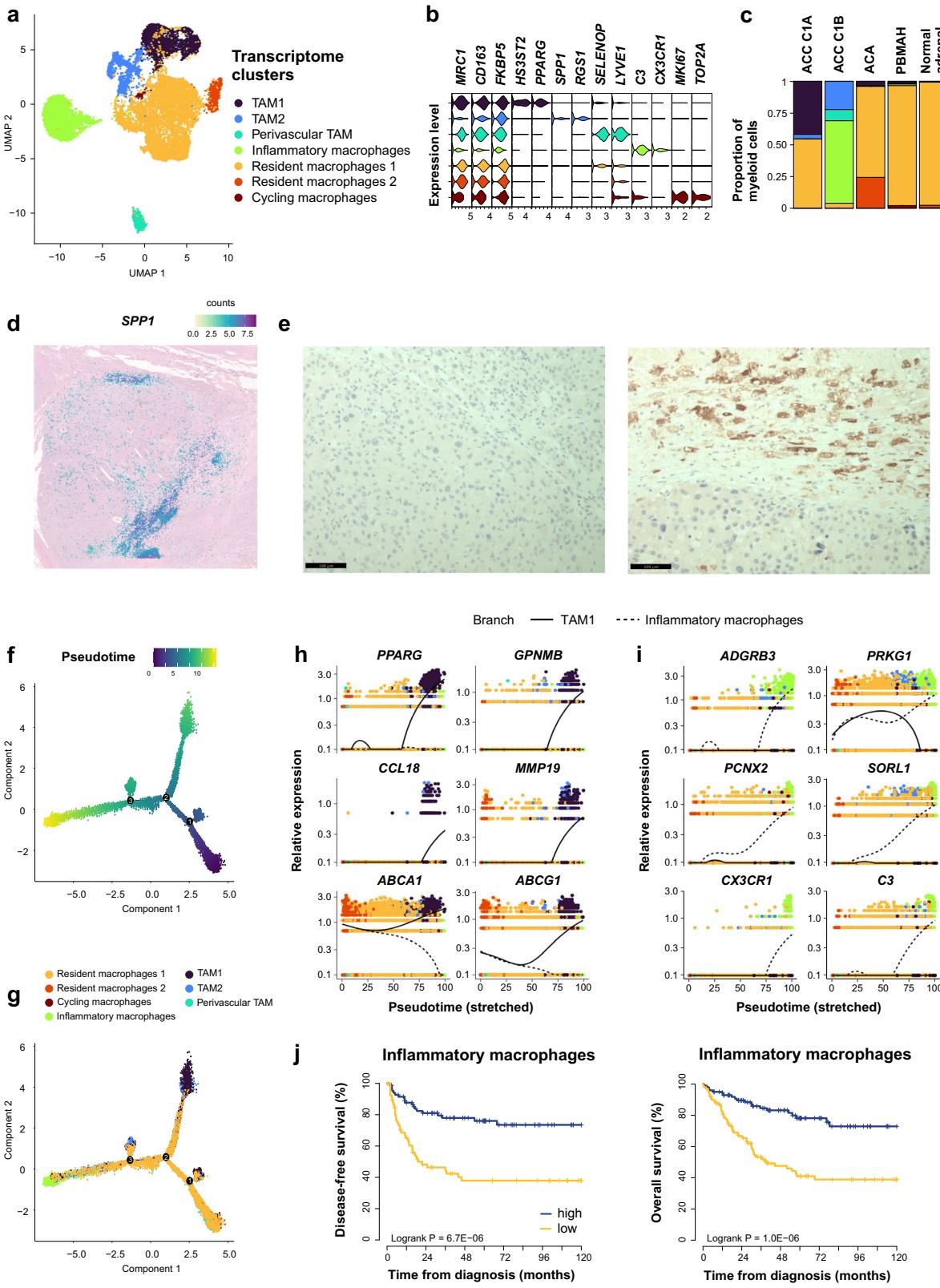

cells in pseudotime trajectory and spatial transcriptomics. This HSP+ signature may reflect a stress response to steroidogenesis-associated reactive oxygen species production[43,44]. Potentially in line with hypothesis, the inference of chromosome alterations with InferCNV in HSP+ cells showed a slight but limited increase of chromosome alterations (Supplementary Fig. 2f).

The functional zonation of normal adrenal gland is also retrieved in tumor steroid cells, with GM1_ZG, GM2_ZF1, GM3_ZF2, and GM4_ZR gene modules reflecting zona glomerulosa, zona fasciculata, and zona reticularis respectively. The co-existence of these different steroid cells signatures within a single tumor questions a potential cell trans-differentiation in tumors. In normal adrenals, trans-differentiation

**Fig. 7 | Characterization of adrenocortical myeloid cells.** Individual myeloid cells from 34 adrenocortical tumors and 4 normal adrenals are presented (single-nucleus transcriptomes). **a** UMAP annotated by transcriptome clusters. **b** Stacked violin plot of transcriptome cluster markers, showing log-scaled raw counts for top differentially expressed genes for each cluster. **c** Proportion of transcriptome clusters in each sample type. **d** High definition spatial transcriptomics of an ACC, expressing the TAM marker *SPP1*. Capture area 6.5 × 6.5 mm.
**e** Immunohistochemistry staining in two ACC, one negative (left) and one positive (right) for SPP1, showing SPP1 staining in intra-tumor macrophages in contact with necrosis. Magnification x400, barscale 100 μm. **f** Myeloid cells trajectory annotated by pseudotime, using discriminative dimensionality reduction via learning a tree (DDRtree). **g** Myeloid cells trajectory annotated by transcriptome clusters, using

DDRtree. **h** Expression profiles of top genes with pseudotime variation from resident macrophages towards TAM1 (solid line), including M2/TAM markers (*PPARG, GPNMB, CCL18*), and anti-inflammatory markers including cholesterol transporters (*ABCA1, ABCG1*) and the metalloproteinase *MMP19*. Black lines: mean expression over pseudotime. **i** Expression profiles of top genes with pseudotime variation from resident macrophages towards inflammatory macrophages (dashed line), including pro-inflammatory markers (*CX3CR1, C3, PRKG1*) and the adhesion G protein-coupled receptor *ADGRB3*. Black lines: mean expression over pseudotime.
**j** Association of inflammatory macrophages signature and outcome. Inflammatory macrophage signature was deconvoluted in bulk ACC transcriptomes from 201 patients with CIBERSORTx. Kaplan–Meier curves represent disease-free and overall survival.

from ZG towards ZF/ZR is related to transient Wnt-βcatenin pathway activation in ZG, followed by cAMP/PKA pathway activation in ZF/ZR[1]. In contrast, in ACC, the Wnt-βcatenin pathway is commonly activated by *CTNNB1* mutations, leading to a constitutive activation that cannot be switched off. In addition, the co-existence of different steroid cells signatures in ACC does not seem to be linked to "classical" tumor sub-clonality, as assessed by chromosomal alterations. Indeed, ACC commonly exhibit chromosomal alterations with >90% clonality[6,7]. This suggests that the observed heterogeneity might instead be due to a functional cellular process leading to different cell states. One hypothesis could be the interplay between the cell cycle and steroidogenesis[45]. Further studies are needed to elucidate the mechanisms of steroid differentiation dynamics in ACC.

The poor outcome associated with GM1_ZG, GM2_ZF1, and GM3_ZF2 compared to GM4_ZR may reflect the pejorative impact of Wnt-βcatenin activation and glucocorticoid (cortisol) secretion on immune microenvironment. Wnt-βcatenin activation in tumor cells results indeed in T-cell exclusion[46], whereas glucocorticoids are major inhibitors of the immune system, preventing T-cell activation and inducing macrophages polarization towards M2/TAM states[47–49].

Our findings support the immunosuppressive effects of cancer cells specifically in ACC, through specific associations of steroid and immune signatures. While single-nucleus isolation could favor the isolation of certain cell types and bias cell proportions, our results are consistent with previous studies identifying ACC as "immune cold" tumors[10,50], with lymphocyte depletion. Despite this "cold" immune landscape, this study demonstrates immune variability in ACC, with different levels of exhausted T-cells, pro-inflammatory "M1-like" macrophages and TAM. ACC steroid and immune single-nucleus signatures combine into distinct tumor ecotypes. Eco2 ZF-like ecotype combines GM1_ZG, GM2_ZF1, and GM3_ZF2 steroid signatures with exhausted T-cell signature, and is associated with poor survival. Conversely, Eco5 ZR-like ecotype combines GM4_ZR and inflammatory macrophages signatures, and is associated with better survival. These two ecotypes elucidate at the single-nucleus level the C1A/C1B bulk transcriptome signatures, with Eco2 associated with C1A and Eco5 associated with C1B. In summary, these ecotypes describe the specific association of steroid cells profiles with immune cell states. The functional importance of tumor immune microenvironment is now well established in response to cytotoxic chemotherapies or immunotherapies in several cancer types[51,52]. Indeed, cytotoxic chemotherapies induce the release of tumor antigens which recruits and activates antigen-presenting cells, ultimately triggering an antitumor adaptive immune response. And immunotherapies boost the antitumor immune response. To which extent these general oncologic mechanisms apply to ACC remains to be established.

Beyond specific steroid and immune cells signatures, this study also demonstrates the existence of cancer-associated signatures in endothelial cells (TEC) and fibroblasts (CAF). TEC2 and CAF3 signatures combine with GM7_Mitosis and GM8_Hypoxia cancer cell signatures into another ecotype, Eco1 hypoxia/mitosis. Eco1 is also

associated with C1A and poor outcome. Eco1 further reflects the mechanisms of aggressiveness in C1A, associating cell proliferation and angiogenesis activation, as seen in a majority of aggressive cancers[53].

ACC ecotypes can be leveraged in a translational framework for patient stratification. Cell states provide a valuable prognostic information, applicable to individual patients. A theranostic perspective could also emerge from the association of cell states and response to treatment, which remains to be explored. Determination of cell states statuses could rely on specific markers or on transcriptomic signatures, that can be inferred from bulk transcriptome. This latter approach is increasingly integrated into routine oncology[54,55] and is applicable to paraffin-embedded samples[22]. Treatment options in advanced ACC are limited, with less than 20% of objective response to mitotane, platinum-based chemotherapy or immune checkpoints inhibitors[56,57]. The single-nucleus atlas of ACC paves the way towards novel therapeutic strategies. Firstly, this study identifies several ligand-receptor interactions implied in immune inhibition and angiogenesis. Of note, these interactions are based only on gene expression levels at this stage and are therefore speculative. However, the list of potential interactions may orient future experimental studies in search for novel therapeutic targets. Secondly, the duality of Eco2/Eco5 steroid cells and immune microenvironment raises the question of a possible reversion of Eco2 ZF-like/exhausted T-cells towards Eco5 ZR-like/inflammatory macrophages. Would it be possible to inhibit gluco-corticoid (cortisol) secretion at the tumor tissue level in ACC with Eco2 ecotype? Novel anticortisolic drugs[58] and combination of these drugs[59] may be sufficient to reach full blockade of intra-tumor cortisol synthesis. Tumor biopsies after treatment would be needed to ascertain the proper intra-tumor cortisol suppression[52]. If full cortisol blockade could be achieved, would it promote the recruitment of inflammatory macrophages as in Eco5 ZR-like? In addition, the Eco5 ZR-like ecotype is potentially signing an androgen secretion at the tissue level. Indeed, the Eco5 GM4_ZR cancer cell signature is enriched in markers of zona reticularis, the functional zone producing androgens in normal adrenals. In recent ACC mouse models, the protective role of androgens has been established through the promotion of phagocytic macrophages[60,61]. However, direct evidence of intra-tumor androgen levels in ACC with Eco5 ZR-like is lacking so far. And whether increasing androgens at the tissue level would inhibit tumor progression through macrophages promotion, in mice and in humans, remains to be established.

Beyond adrenal tumors, modulation of immune microenvironment by intra-tumor glucocorticoids has been suggested. Indeed, recent studies report evidence of intra-tumor glucocorticoids in other tumor types, either by de novo synthesis[48] or by conversion of cortisone to cortisol[62]. Furthermore, enzymatic inhibition of cortisone conversion promotes an anti-tumor microenvironment[62]. To which extent modulation of intra-tumor glucocorticoids - and potentially of intra-tumor androgens - would reverse immune escape and promote response to immune checkpoints inhibitors remains to be established.

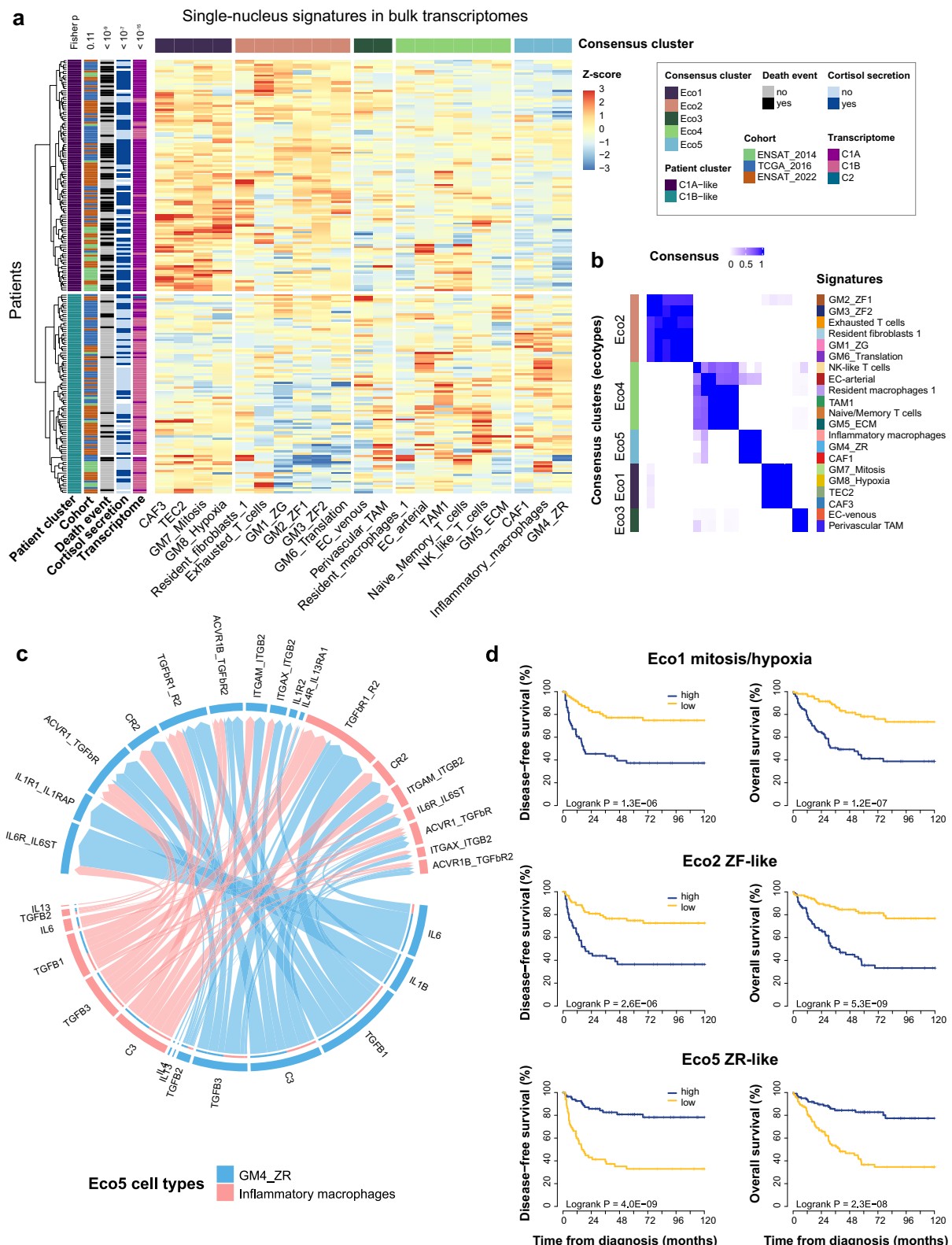

**Fig. 8 | Interactions between single-nucleus signatures. a** Hierarchical clustering of the main single-nucleus signatures in the 201 ACC bulk transcriptomes. The single-nucleus signatures included 13 main microenvironment signatures and 8 gene modules scores. These signatures were transformed into Z-scores in each of the three ACC bulk datasets (ENSAT_2014, TCGA_2016, ENSAT_2022) before clustering. Eco1-5 represent the different ecotypes. **b** Consensus partition clustering of the main single-nucleus signatures in ACC bulk transcriptomes. **c** Chord plot representation of main ligand-receptor interactions related to inflammation and

immune activation in Eco5 ecotype. Ligand-receptor pairs interactions related to inflammation and immune activation are presented for the following signaling pathways enriched in Eco5: FASLG, LT, IL4, IL6, MSTN, TGFb, COMPLEMENT, CD40, CD80, CD86, ICAM, CCL, CXCL, and IL1. **d** Association of ecotypes and outcome. Ecotype scores (sum of single-nucleus signatures scores in each ecotype) were computed in bulk ACC transcriptomes from 201 patients. Kaplan–Meier curves represent disease-free and overall survival.

In this perspective, adrenal immune cells profiles may be used for evaluating the impact of intra-tumor steroids in cancers.

In conclusion, single-nucleus transcriptome identifies the dynamics of steroid cell differentiation in normal adrenals and adrenocortical tumors, with an intermediate differentiation state and a parallel between tumorigenesis and adrenal functional zonation. Adrenal cortex microenvironment is scarce but heterogeneous, with distinct immune and stromal signatures associated with ACC outcome. This atlas opens potential perspectives for the treatment of advanced tumors.

## Methods

### Experimental model and participants' details

**Ethical statement.** This study was reviewed and approved by the Institutional Review Board "Comité de protection des personnes Ile de France 1" (application #13311 COMETE-TACTIC). Signed informed consent for somatic molecular analysis and for access to de-identified clinical data was obtained from each patient.

Patients did not receive any compensation for the study.

**Human tumor samples.** A total of 39 adrenal samples from 36 patients were included. Age, gender, tumor type, and other clinical features are provided in Supplementary Data 1 and 2. Gender was determined based on self-report and was considered in the study design to reflect the sex ratios typically observed in the different types of adrenal tumors. All but one patient (ACC17) were operated at Cochin Endocrine Surgery Department.

Adrenal samples were snap-frozen into liquid nitrogen immediately after surgery, then stored at −80 °C. Thirty-eight samples from 35 patients (28 females/7 males, median age 46 years) were used for single-nucleus RNA-seq. For three patients, two tumor samples were collected: (i) one primary tumor (ACC1a) and one metastasis (ACC1b); (ii) two distinct regions from a primary tumor, one with aggressive morphological features (ACC2b and ACC8a) and the other without (ACC2a and ACC8b).

One frozen sample (ACC2) was used for spatial transcriptomics.

Remaining tissue samples were formalin-fixed, and paraffin embedded (FFPE) for histological examination and immunohistochemical study by adrenal expert pathologists (MS, FV). Two FFPE samples (NAd4 and NAd5) were used for spatial transcriptomics.

Extensive hormone and imaging explorations were performed at diagnosis following standard adrenal tumors guidelines[63,64]. After surgery, additional treatments and follow-up were performed following the clinical guidelines[63,64]. All samples were collected before the initiation of antitumor treatment.

**Public datasets.** Three bulk transcriptome cohorts of patients with ACC were used to test the association between single-nucleus signatures and outcome. These cohorts, referred to as ENSAT_2014, TCGA_2016, and ENSAT_2022, were generated with Affymetrix U133_Plus_2.0 microarray from frozen samples, full-length RNA-sequencing from frozen samples, and 3′ RNA-sequencing from FFPE samples respectively[6,7,22].

### Single-nuclei RNA sequencing data production, processing and analysis

**Tissue dissociation and isolation of single nuclei.** Single nuclei were isolated using an in-house protocol[65]. Frozen tissue samples were minced in a lysis buffer (10 mM Tris–HCl, 10 mM NaCl, 3 mM MgCl2 and 0,1% Nonidet TM P40 in Nuclease Free-Water) and mechanically dissociated using a A and B pestle gently 15 times each. Samples were then suspended in 2% BSA PBS, sieved through 100 μm cell strainers (VWR), centrifuged twice for 10 min at 500 g with 2% BSA PBS resuspension. Nuclei were stained with the Alexa Fluor® 647-conjugated Mab414 antibody (BioLegend, clone Mab414, 1:250) targeting Nuclear

Pore Complex proteins. Sorting was performed on a FACSAria™ III cell sorter (BD Biosciences) equipped with an 85 μm nozzle using BD FACSDiva™ software. The gating strategy consisted of selecting nuclei based on their size and internal complexity on a FSC-A vs. SSC-A dot plot to exclude debris and isolating Mab414-positive nuclei on a FSC-A vs. Alexa Fluor® 647 fluorescence dot plot (Supplementary Fig. 1a). Sorted nuclei were immediately processed on a Chromium Controller (10x Genomics).

**Droplet-based snRNA-seq.** Single-nucleus RNA sequencing was performed using the Chromium Single-Cell v3 3′ Gel Bead, Chip, and Library Kits (10x Genomics) according to the manufacturer's protocol. A total of 5000 cells were targeted per bead. Libraries were sequenced on a NextSeq 500 platform (Illumina) with paired-end sequencing. A total of 28, 8 and 56 cycles were run for Read 1, i7 index and Read 2, respectively.

**SnRNA-seq data processing, cluster annotation and data integration.** Raw BCL files were demultiplexed and mapped to the reference genome GRCh38, including pre-mature mRNA sequences, using the Cell Ranger Single Cell v3.1.0 software (10x Genomics). Doublets were filtered out using Scrublet[66] v0.2.3 with default parameters.

We used the Seurat[67] v4.3.0 in R v4.3.1 for subsequent analyses. To filter out low-quality nuclei, nuclei with <500 detected genes, >8000 detected genes, or >5% of mitochondrial transcripts were removed. Data were normalized using SCTransform method[68], including a regression step on mitochondrial transcripts if they impacted the clustering (normal and tumor steroid cells, fibroblasts, endothelial cells, lymphocytes). Dimensionality reduction was performed using principal component analysis (PCA) depending on the elbow plot inflexion point, with 50 dimensions for general cell atlas, 5 dimensions for lymphoid cells, and 10 dimensions for all other analyses. Clustering was performed using graph-based clustering (*FindNeighbors* and *FindClusters* functions), and visualized using Uniform Manifold Approximation and Projection (UMAP). Clustering resolution was selected on clusters stability using Clustree[69] v0.5.0. To achieve the best balance between over- and under-clustering, resolution was chosen based on the following criteria: (i) clusters remain consistent across small resolution changes, (ii) separate groups of cells on UMAP form distinct clusters, and (iii) each cluster shows differentially expressed genes compared to others.

Cells were annotated with SciBet[70] v1.0 and Garnett[71] v0.1.23, using the following references: a melanoma dataset for microenvironment[11] and a fetal adrenal dataset for adrenocortical cells[18]. Cells properly clustered and annotated were further processed for subclustering.

For the global cell atlas and microenvironment analyses, no batch correction was applied, since microenvironment cells properly clustered irrespective of their sample of origin. For normal adrenal steroid cells, data integration was performed to overcome inter-individual variability. We used SCTransform normalization and canonical correlation analysis with default parameters. For tumor steroid cells, data integration tools were tested to overcome inter-individual variability, but led to over-correction. We therefore used another method (recurrent gene modules; see below) to explore intra-tumor variability of tumor steroid cells.

**Inferred CNV analysis from snRNA-seq.** The copy number variation (CNV) signal for steroid cells was estimated using the InferCNV method[19] with a 100-gene sliding window. Genes with a mean count of less than 0.1 across all cells were filtered out before the analysis and the signal was denoised using a threshold of 1.5 standard deviation from the mean. For each window, CNV were inferred if the CNV signal was out of the 99% confidence interval of the CNV signal distribution in steroid cells from normal adrenals.

**Differential gene expression analyses.** For each cluster, a list of differentially expressed genes (DEG) was established in comparison with all other clusters. Group comparisons were performed on raw counts, using the Wilcoxon rank-sum test with Bonferroni correction, implemented in Seurat FindAllMarkers function, with the following parameters: filtering only positive differences, with log2 fold-change threshold >0.25, and genes expressed in >10% of cells in one group.

**Gene set enrichment analysis.** For each cluster, the top 100 most differentially expressed genes with adjusted $p < 0.05$ were selected for enrichment analyses. Over-representation of these genes in human MSigDB collections (Gene Ontology-Biological Process (GO-BP) and Reactome databases) was tested using the ClusterProfiler[72] package v4.10.0 (enricher, enrichGO and compareCluster functions).

**Pseudotime trajectory analysis.** Cells were ordered in pseudotime trajectories with Monocle[73] v2.30.0, using default parameters. The 1000 most variable genes were used for cell ordering. Dimensionality reduction was performed with DDRTree without additional normalization. For normal steroid cells, zona glomerulosa was defined as the starting point of the trajectory. For microenvironment cells, the most abundant cluster in normal adrenal samples was defined as the starting point of the trajectory. Genes significantly varying with pseudotime were identified with the differentialGeneTest function for linear trajectories, and with the BEAM function for branch trajectories.

**Gene modules of intra-tumor heterogeneity.** Recurrent gene modules were defined by the following steps: (i) Capturing gene modules with variable expression among steroid cells in each tumor. For each tumor, gene expression variability was captured by the 10 first PCA components. For each component, two gene modules were selected, corresponding to the top 50 positively and top 50 negatively associated genes; (ii) Filtering out non-recurrent gene modules. Among the different tumors, gene modules similarity was computed with Sørensen index (S(A ∩ B) = 2* (A ∩ B) / (A + B)). The similarity matrix was filtered to discard gene modules associated with <2 other gene modules (Sørensen indexes <0.4); (iii) Aggregating gene modules into clusters of recurrent gene modules. Hierarchical clustering of the filtered similarity matrix was performed. The corresponding dendrogram was cut to obtain clusters of gene modules coming from ⩾3 tumors; (iv) Filtering genes for each recurrent gene module. In each cluster, genes contributing to gene modules found in ⩾50% of tumors were selected. Of note, two tumors (ACC3, ACC14) were excluded due to the limited number of steroid cells (<250).

**Cell-cell interactions in single-nucleus transcriptomes.** Cell-cell interactions were explored with Cellchat[40] v1.6.1, with the following steps: (i) Single cells selection and labeling: steroid cells were labeled with the gene module with the highest score. Microenvironment cells from the 13 clusters used for deconvolution in bulk transcriptomes were selected; (ii) Computation of cell-cell interaction probability for all possible ligand-receptor pairs in the Cellchat database (computeCommunProb function); (iii) Integration of ligand-receptor interactions into signaling pathways (computeCommunProbPathway function); (iv) Selection of the signaling pathways with high cell-cell interaction signals: for each signaling pathway, interaction probabilities were transformed into Z-scores. Pathways with at least one highly probable interaction (Z-score > 3) were selected; (v) Selection of interactions between cells belonging to the same ecotype.

The relationship between signaling pathways was explored by unsupervised hierarchical clustering of interaction scores.

## Bulk transcriptomes analysis

**Single-nucleus signatures in bulk transcriptomes.** In bulk transcriptomes, single-nucleus signatures were explored in two ways:

- For gene modules reflecting steroid cells heterogeneity, gene set enrichment scores were computed in each sample with single sample GSEA[74] (GSVA package v1.36.2).
- For microenvironment cells transcriptome profiles, signature deconvolution was performed using the CIBERSORTx Cell Fractions module[75], with the parameters recommended for 10x single-cell data (single_cell, fraction 0.1, rmbatchSmode). Thirteen out of 27 signatures were considered, discarding 14 signatures with low number of cells (<50 in ACC), or with intermediate signatures in trajectories (CAF2, TEC1, TAM2). To evaluate the reliability of deconvolution for resolving each cell subset in bulk tissue, we randomly sampled each population in 1000 cells (if >2000 cells) or 50% of cells (if <2000 cells) for training set, used to create the reference matrix for deconvolution, and the remaining cells for testing set, used to create a pseudo-bulk dataset. Cell proportions estimated by deconvolution in snRNA-seq pseudobulk and real cell proportions were highly correlated ($r = 0.71$).

These signatures were transformed into Z-scores to merge the three ACC bulk dataset, and capped within the −3: + 3 range before clustering.

**Aggregation of single-nucleus signatures into ecotypes.** In bulk transcriptomes from 201 patients, single-nucleus signatures scores were clustered using unsupervised hierarchical clustering. Ecotypes were defined as groups of single-nucleus signatures, following consensus partition clustering (cola package[76] v2.8.0, consensus_partition, select_partition_number and consensus_heatmap functions with default parameters except max_k = 10).

Ecotype scores were computed for each patient, by adding the scores of single- nucleus signatures corresponding to each ecotype.

## Spatial transcriptomics data production, processing and analysis

**Spatial transcriptomics.** Tissue samples were cut into 10-µm sections. One frozen and two FFPE samples (3 females, median age 66 years) were processed using the Visium Spatial Gene Expression and Visium Spatial for FFPE Gene Expression Kits respectively (10x Genomics) according to the manufacturer's protocol.

For the frozen sample, adrenal tissue permeabilization condition was optimized using the Visium Spatial Tissue Optimization Kit (10x Genomics), which was found to be ideal at 18 min.

Sections were stained with H&E (Haematoxylin and eosin, Dako) and imaged using a Lamina Slide Scanner (Perkin Elmer), then processed for spatial transcriptomics. The resulting complementary DNA library was amplified (17 cycles for the frozen sample, 14 cycles for FFPE samples) using the Kappa Sybr Fast qPCR kit (Kappa biosystems). Dual indexed libraries were prepared using the Library Construction Kit with the Dual Index Kit TT Set A (10x Genomics) for the frozen sample, and with the Dual Index TS Set A kit (10x Genomics) for FFPE samples, according to the manufacturer's protocol. Paired end dual indexed sequencing was performed on a NextSeq 500 platform (Illumina).

High definition spatial transcriptomics was performed for one ACC sample following the Visium HD Spatial Gene Expression protocol (10x Genomics).

**Spatial transcriptomics data processing and clustering.** Raw BCL files were demultiplexed and mapped to either the human GRCh38 genome assembly for the frozen sample, or to the Visium Human Transcriptome Probe Set V2.0 GRCh38 for the FFPE samples, using the Space Ranger Single Cell v1.3.1 software (10x Genomics).

We used the Seurat[67] v4.3.0 in R v4.3.1 for subsequent analyses. Data were normalized using SCTransform method[68], including a regression step on mitochondrial transcripts for the frozen sample. Dimensionality reduction was performed using the first 30 dimensions of principal component analysis (PCA). Clustering was performed using graph-based clustering (FindNeighbors and FindClusters functions), and visualized using Uniform Manifold Approximation and Projection (UMAP). Clustering resolution was selected on clusters stability using Clustree[69] v0.5.0.

Data integration was performed for the analysis of FFPE samples with SCTransform normalization and reciprocal PCA (RPCA) with default parameters.

**Inferred CNV analysis from spatial transcriptomics.** Inference of chromosome alterations was performed with InferCNV[19], using a benign tumor as a reference (also processed with Visium Spatial Gene Expression). Inference was performed in ACC2 and in a benign tumor (control).

**Single-nucleus signatures in spatial transcriptomes.** In spatial transcriptomes, single-nucleus signatures were explored in two ways:

- For gene modules reflecting steroid cells heterogeneity, gene set enrichment scores were computed in each spot with the Seurat AddModuleScore function.
- For normal steroid cells and microenvironment cells transcriptome profiles, signature deconvolution was performed with Cell2location[77] v0.1.3, using an hyper-prior of $n = 5$ cells by spot.

**Immunohistochemistry analyses**
Immunohistochemistries were performed on dewaxed 3 μm slides using a leica BOND III device (Leica, Berlin, Germany). Double staining with anti-DAB2 (clone HPA028888, 1:800, Sigma Aldricht, Saint Louis, USA) and anti-CYP17A1 (clone HPA048533, 1:800, Sigma Aldricht, Saint Louis, USA) antibodies was carried out with a pH6 buffer solution (EDTA buffer, Bond Epitope Retrieval Solution 1, Leica, Berlin, Germany) for 20 min by antibody. Single staining with anti-SPP1 (clone HPA027541, 1: 475, Sigma Aldricht, Saint Louis, USA) antibody was performed with a pH9 buffer solution (EDTA buffer, Bond Epitope Retrieval Solution 2, Leica, Berlin, Germany) for 20 min.

**Statistical analyses**
Statistical analysis was performed using R (version 4.3.1, R Stats, survival, survcomp).

Comparisons between groups were performed with the Student's $t$ test for normally distributed quantitative variables, with the Wilcoxon and Kruskal-Wallis tests for not normally distributed quantitative variables, and with Fisher's test for qualitative variables. For large contingency tables, the p-values were estimated using a Monte Carlo simulation approach, with 2000 replicates.

Correlations were computed with Pearson's coefficient for normally distributed quantitative variables, and with Spearman's coefficient for not normally distributed quantitative variables and ordinal variables.

Disease-free survival (DFS) was analyzed in stage I–III ACC, and overall survival (OS) was analyzed in stage I–IV ACC. Survival curves of high and low signatures scores (>vs <median) were obtained with Kaplan–Meier estimates and compared with the log-rank test. Cox proportional hazards regression was used to identify variables associated with DFS and OS. Significant variables were combined into stepwise multivariable models. To evaluate the value of single-cell signatures compared to existing ACC prognostic models, we computed C-indexes for DFS and OS models based on existing prognostic factors alone and along with single-cell signatures. Comparison of nested models was performed using likelihood-ratio test.

Adjustments for multiple testing were performed using Benjamini-Hochberg method.

All tests were two-sided unless otherwise specified, and the level of significance was set at $p < 0.05$.

**Reporting summary**
Further information on research design is available in the Nature Portfolio Reporting Summary linked to this article.

## Data availability
The raw single-nucleus RNA-sequencing and spatial transcriptomics data generated in this study have been anonymized using BAMboozle[78] and deposited in the European Genome-phenome Archive (EGA) database under accession code EGAD50000000835. Counts tables of single-nucleus RNA-sequencing have been deposited on EGA under accession code EGAD50000000836 and on Zenodo[79] [https://doi.org/10.5281/zenodo.10534061] (ACC) and [https://doi.org/10.5281/zenodo.10534245] (benign ACT and normal adrenals). Counts tables and images of spatial transcriptomics have been deposited on EGA under accession code EGAD50000000836 and on Zenodo[79] [https://doi.org/10.5281/zenodo.10560206] (ACC) and [https://doi.org/10.5281/zenodo.10560525] normal adrenals). The public transcriptome datasets used in this study are openly available and can be accessed as follows: - "ENSAT_2014" in the Gene Expression Omnibus (GEO) repository under accession number GSE49280. - "TCGA_2016" in the GDC portal [https://portal.gdc.cancer.gov/projects/TCGA-ACC] - "ENSAT_2022" in the supplementary tables of the original paper[22] [https://academic.oup.com/ejendo/article/186/6/607/6853696#supplementary-data].

## Code availability
Code related to the analyses in this study mainly reuse codes from other softwares (Cell Ranger, Space Ranger, Scrublet, Seurat, Clustree, SciBet, Garnett, InferCNV, ClusterProfiler, Monocle, GSVA, CIBERSORTx, cola, CellChat, Cell2location) and can be found on GitHub at https://github.com/GESTE-IC/snRNAseq_ACT_atlas[80] under MIT License.

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

## Acknowledgements

This work was supported by the SIRIC CARPEM (CAncer Research for PErsonalized Medicine, "2020 Emergence and Innovation" grant to G.A. and F.L.), the ITMO Cancer AVIESAN (Alliance Nationale pour les Sciences de la Vie et de la Santé, National Alliance for Life Sciences & Health, "2018 Single-Cell" grant to G.A., "2017 FRFT" PhD grant to A.J., "MIC 2022" grant to G.A.) within the framework of the Cancer Plan, the Agence Nationale de la Recherche (ANR-18-CE14-0008-01 to J.B., ANR-24-CE14-3550 to A.J.), the Fondation pour la Recherche Médicale (FRM EQU201903007854 to J.B.), the Pro-gramme de Recherche Translationnelle en Cancérologie (PRT-K COMETE-TACTIC 148663 to J.B. and G.A. and PRT-K 2020 COMETE-CARE to G.A. and A.J.) the Fondation ARC ("2021PJA3" grant to G.A.), and COST Action CA20122 Harmonization (Travel Grant to A.J.). The collection of tumors was sponsored by Assistance Publique – Hôpitaux de Paris (Délégation à la Recherche Clinique et à l'Inno-vation, AP-HP DRCI). We thank the GENOMIC, CYBIO and HISTIM platforms, the team "Genomic and Signaling of Endocrine Tumors" of Institut Cochin, Violaine Emourgeon and Lauren Demerville (AP-HP DRCI), and the French COMETE and European ENSAT research networks. Endocrine department is part of the European Reference Network on Rare Endocrine Conditions (Endo-ERN) – Project ID No 739572 and the European Reference Network EURACAN.

## Author contributions

Conceptualization, A.J., J.B., and G.A.; Methodology, A.J., Y.M., T.F., Y.B., P.C., C.d.G., A.F., D.d.M., R.A., N.B., and G.A.; Investigation, A.J., F.V., B.I., F.L., C.B., M.A., R.O., M.F., M.F.B., and K.P.; Resources, A.J., F.V., M.S., P.V., L.B., F.B.S., B.D., M.G., E.P., M.B., A.D., M.H., A.T., R.L., L.Gr., L.Gu., A.B., B.R., J.B., and G.A.; Writing – Original Draft, A.J., Y.M., and G.A.; Writing – Review & Editing, all authors; Supervision, A.J., J.B., and G.A.; Funding Acquisition, A.J., J.B., and G.A.

## Competing interests

The authors declare no competing interests.

## Additional information

[1]Université Paris Cité, CNRS, Inserm, Institut Cochin, Paris, France. [2]Department of Endocrinology and National Reference Center for Rare Adrenal Disorders, AP-HP, Hôpital Cochin, Paris, France. [3]Department of Pathology, AP-HP, Hôpital Cochin, Paris, France. [4]ENS Paris-Saclay, Gif-sur-Yvette, France. [5]Department of Hormonology, AP-HP, Hôpital Cochin, Paris, France. [6]Department of Digestive and Endocrine Surgery, AP-HP, Hôpital Cochin, Paris, France. [7]Department of Genetics, Fédération de Génétique et Médecine Génomique, AP-HP, Hôpital Cochin, Paris, France. [8]Department of Radiology, AP-HP, Hôpital Cochin, Paris, France. [9]Department of Endocrinology, diabetes and nutrition, University of Bordeaux, CHU Haut Leveque, Pessac, France.
✉e-mail: anne.jouinot@inserm.fr; guillaume.assie@aphp.fr

