## [Peer Review File · Nature Communications]

Impact of steroid differentiation on tumor microenvironment revealed by single-nucleus atlas of adrenal tumors

Corresponding Author: Dr Anne Jouinot

Version 0:

Reviewer comments:

Reviewer #1

(Remarks to the Author)

The manuscript, titled "Impact of steroid differentiation on tumor microenvironment revealed by single-nucleus atlas of adrenal tumors," offers valuable insights into the differentiation of steroid cells and their interaction with the tumor microenvironment in adrenal tumors, particularly adrenocortical carcinomas (ACC). The single-nucleus transcriptome atlas provided is comprehensive, showcasing the strengths of integrating normal and tumoral adrenal tissue data. The combination of spatial transcriptomics and pseudotime trajectory analysis adds robustness to the findings. However, despite the strengths, several areas require more clarity, especially regarding clinical relevance and the robustness of the analytical methods. Overall, the study has merit but requires substantial revision.

Major comments:

1. The entire article lacks experimental verification. Even for the identified cell populations, there is no experimental verification of marker genes expression. It is recommended to add experimental verification such as IHC staining in patient specimens to further consolidate the observations and conclusions of the manuscript.
2. Figure 3c looks exactly the same as Figure 1a, so Figure 3c shows tumor cells or total cells? I think if the author wants to concentrate only on tumor cells in this section, tumor cells should be extracted, re-classified and re-annotated to avoid interference from other normal cells.
3. In figure 2i, the number of cells in the intermediate state appears to be significantly less than that in the figure 2a. Please explain why spatial transcriptome sequencing captures fewer intermediate cells than single-nucleus sequencing. Is it a technical problem or something else?
4. While the spatial transcriptomics data adds valuable spatial context to the atlas, the manuscript could benefit from deeper integration of this data with the single-nucleus transcriptomic findings. Currently, the spatial analysis appears secondary to the primary single-nucleus data. The authors should more thoroughly integrate these data with the single-nucleus findings, particularly by showing how spatial heterogeneity may correspond to functional or clinical differences.
5. The manuscript introduces five distinct ecotypes based on single-nucleus signatures and suggests that these correlate with clinical outcomes. However, it is not clear how these ecotypes were validated externally, particularly across different cohorts. The use of bulk transcriptomic datasets is valuable, but more detail is needed on how well these bulk signatures recapitulate single-nucleus findings. Additionally, it would strengthen the manuscript to include more direct comparisons to existing ACC prognostic models to clarify the incremental value of the ecotypes.
6. While the identification of ecotypes is novel, the manuscript does not provide enough detail on how the identified ecotypes or gene modules could be applied in clinical practice. The manuscript should clearly outline how these findings could directly influence therapeutic strategies, especially regarding the feasibility of manipulating steroidogenesis and immune cell interactions. Currently, the discussion on therapeutic implications is somewhat speculative and lacks sufficient grounding in experimental data.
7. The manuscript proposes that modulation of glucocorticoid and androgen levels in tumors could influence immune microenvironments and thus patient outcomes. While intriguing, the proposed mechanism is presented without sufficient experimental validation. The authors should either provide additional evidence supporting this claim or moderate their conclusions to avoid overinterpreting the data.
8. The use of single-nucleus sequencing is commendable and appropriate for the study's objectives. However, the manuscript lacks detail in the description of how quality control was ensured, particularly in terms of potential biases introduced during single-nucleus isolation. Additionally, more transparency is required in reporting batch effects and how these were controlled, especially for the clustering analysis (UMAP). There is mention of the clustering stability, but the

clustering resolution decisions should be more thoroughly explained and justified.

Minor comments:

1. In figure 1f, the cells in gray should also be labeled.
2. In line 181, some “-” (hyphen) are before the words, some are after the words, and the usage is confusing. It can be used more clearly.
3. Figure 8a, two strips (Death event, Cortisol secretion) are in gray and black. It looks a little bit confusing.
4. The study's identification of immune cell signatures, such as exhausted T cells and inflammatory macrophages, is a critical finding. However, the functional role of these immune cell types in the context of ACC needs further exploration. It is suggested that the authors either include additional functional assays or cite more comprehensive literature to support their conclusions on immune cell behavior and its potential impact on prognosis.
5. The discussion should more clearly articulate how these findings could be translated into clinical practice. Specifically, the practical steps for using these ecotypes or gene modules as biomarkers for patient stratification should be better outlined.

Reviewer #2

(Remarks to the Author)

In this study, Jouinot et al present a single-nucleus transcriptome atlas of 38 human adrenal cortex samples, covering diverse tumor and microenvironment cell types from primary ACCs, ACAs, PBMAHs, and normal adrenals. The dataset is further complemented by spatial transcriptomic profiling of normal and tumor adrenal samples. This atlas enabled the authors to investigate differentiation patterns and functional zonations of normal steroid cells, identify recurrent gene expression programs associated with intratumoral heterogeneity among adrenal tumors, and extract novel gene expression/ecotype signatures correlating with survival.

Overall, the dataset generated in this study provides a substantial resource for the molecular characterization of cellular heterogeneity in diverse adrenal tumors. Existing studies employing similar techniques (snRNA-seq and spatial transcriptomics) have largely focused on single adrenal tumor types or normal adrenal tissues only. The analyses applied (trajectory inference, decomposition of gene expression programs, CNV analysis, etc) are standard and mostly appropriate.

1. Existence of a transitional cell state between ZG and ZF (Line 96). The authors claim to identify an intermediate-state cell cluster that shows features of both ZG and ZF, and that the existence of such an intermediate state supports the gradual differentiation pattern from ZG to ZF cells, as observed in mice lineage tracing studies. Such intermediate-state cells appear to be a substantial proportion (~20-25%) of steroid cells in normal adrenals (Fig. 2a,c). This implies that normal steroid cells continuously undergo lineage conversion en masse under homeostasis, which seems unlikely. For example, this population can potentially be explained by differences in clustering resolutions or cell-type annotation compared to previous studies. Are such intermediate-state cells also observed in previously published adrenal snRNA-seq datasets? The authors were able to map such transitional cell cluster onto the spatial data, but in much fewer numbers (Fig. 2i). How do the authors explain such discrepancy in cell proportions? Also, since the Visium data is not single-cell resolution, such intermediate state spots can be explained by mixing of ZG and ZF cells. Overall, there is limited evidence that such a transitional population exists. It would be convincing to validate the existence and numbers of such intermediate-state cells via immunofluorescent staining.
2. Differentiation states of tumor cells that resemble adrenocortical zonations. Via analysis of gene modules extracted using PCA, the authors find that functional zonation of the normal adrenal cortex is also recapitulated in tumor cells (Fig. 3). Is there evidence of lineage conversion between different tumor cell states (ZG/ZF/ZR-like tumor cells)? How dynamic is this process? This can be addressed by looking at the distribution of tumor cell states along subclonal phylogeny reconstructed by CNVs from snRNA-seq.
3. Spatial transcriptomic analysis of ACC2 (Line 221). It is unclear whether all three clusters represent malignant cells instead of normal/stromal cells. What is the CNV status of spots, which may mark tumor margins? Is there a pathology annotation of tissue regions (aggressive features, tumor margin)? Are certain clusters enriched with aggressive features?
4. Validation of cell-cell interactions in tumor ecotypes (Line 243). The authors identified a few interesting ligand-receptor interactions between tumor cells and stromal/immune cells. Can these specific cell-cell interactions be validated in the spatial transcriptomics data?
5. HSP+ cells. The authors identify a distinct steroid cluster with high expression of heat shock proteins (Fig. 2), potentially reflecting a stress response to ROS production associated with steroidogenesis. This is an interesting observation; is there evidence of DNA damage or chromosomal aberrations in such cells? Can the authors speculate whether these cells could be the cell-of-origin of certain adrenal tumors?

Reviewer #3

(Remarks to the Author)

I have reviewed the manuscript by Dr. Jouinot and co-workers, which presents a comprehensive single-cell atlas of the adrenal gland, adrenal tumors and adrenocortical carcinoma. I was impressed by this work that succeeds not only in performing advanced molecular/bioinformatics analyses but also excels in integrating these findings with the current literature in a clear and logical manner. This accomplishment is particularly noteworthy in an era where data is abundant, but refined analyses and well-structured presentations are more scarce. I have only minor comments of how this manuscript may

be improved.

1. Abstract: The abstract has a less refined feeling to it than the text in the full article. It is also unclear to the reader what are novel/validatory findings.

2. Clinical annotations:

a. Were all tumour samples therapy-naïve?

b. Starting at line 206, the manuscript investigates tumour regions exhibiting "aggressive features." It would be beneficial to clearly define what is meant by "aggressive features".

3. Multivariate model: The manuscript reports on the prognostic relevance of "proliferation" and "ZF-like" ecotypes. I'm curious to how do these findings align with those already established in clinical practice, specifically regarding tumour proliferation rate (e.g., Ki-67) and hormone secretion? It would be useful to further discuss the connection between these molecular findings and established clinical markers.

4. Figures:

a. Figure 1: The colour scheme used to represent different cell types can sometimes be challenging to follow, particularly when distinguishing between Chromaffin and Myeloid cells. In panel C, the order in which the cell types appear differs from panels A and B, which may contribute to confusion. It would be helpful to harmonize the presentation across all panels.

b. Figure 2: The legend could be improved to better guide the reader in interpreting the data. For example, I assume that a "dot" represents a "cell" in panels A, D, and F, and that the black line in panel F indicates the mean or median. Explicitly stating these points in the figure legend would enhance the clarity for readers not experts in interpreting data from single cell experiments.

Reviewer #4

(Remarks to the Author)

Reviewer #5

(Remarks to the Author)

Version 1:

Reviewer comments:

Reviewer #1

(Remarks to the Author)

In the revised manuscript, the authors have provided additional experimental evidence to address my previous concerns. These newly added data substantially solidified the conclusions of the manuscript.

(Remarks on code availability)

Reviewer #2

(Remarks to the Author)

The authors have addressed my concerns.

(Remarks on code availability)

The code is complete and well-organized.

Reviewer #3

(Remarks to the Author)

I have no additional comments on the revised manuscript.

(Remarks on code availability)

Reviewer #4

(Remarks to the Author)

(Remarks on code availability)

Reviewer #5

(Remarks to the Author)

(Remarks on code availability)

We thank the Editor and Reviewers for their constructive and detailed comments that were helpful to improve the manuscript. Our point-by-point response is developed below and we provide the following modifications in the revised version of the manuscript.

REVIEWER COMMENTS

Reviewer #1 (Remarks to the Author): Expert in cancer single-cell omics

The manuscript, titled "Impact of steroid differentiation on tumor microenvironment revealed by single-nucleus atlas of adrenal tumors," offers valuable insights into the differentiation of steroid cells and their interaction with the tumor microenvironment in adrenal tumors, particularly adrenocortical carcinomas (ACC). The single-nucleus transcriptome atlas provided is comprehensive, showcasing the strengths of integrating normal and tumoral adrenal tissue data. The combination of spatial transcriptomics and pseudotime trajectory analysis adds robustness to the findings. However, despite the strengths, several areas require more clarity, especially regarding clinical relevance and the robustness of the analytical methods. Overall, the study has merit but requires substantial revision.

Major comments:

1. The entire article lacks experimental verification. Even for the identified cell populations, there is no experimental verification of marker genes expression. It is recommended to add experimental verification such as IHC staining in patient specimens to further consolidate the observations and conclusions of the manuscript.

Following the reviewer's comment, we now included additional experimental validation:

- Morphological validation on adrenal tumor slides by an expert pathologist. Morphological annotations are now provided:

Modified Fig. 2g. Hematoxylin eosin coloration of NAd4 [...]

Modified Extended Data Fig. 2i. Hematoxylin eosin coloration of NAd5 [...]

New Extended Data Fig. 8a. Hematoxylin eosin coloration of the ACC18 sample [...]

Modified Extended Data Fig. 9c. Hematoxylin eosin coloration of the ACC2 sample [...]

- Validation of intermediate state cells in normal adrenals using IHC staining:

New Fig. 2i: DAB2 (ZG marker, brown) and CYP17A1 (ZF/ZR marker, red) immunohistochemistry staining [...]

Page 17 Lines 571-578, Methods section: "*Immunohistochemistry analyses*

Immunohistochemistries were performed on dewaxed 3 μm slides using a leica BOND III device (Leica, Berlin, Germany). Double staining with anti-DAB2 (clone HPA028888, 1:800, Sigma Aldrich, Saint Louis, USA) and anti-CYP17A1 (clone HPA048533, 1:800, Sigma Aldrich, Saint Louis, USA) antibodies was carried out with a pH6 buffer solution (EDTA buffer, Bond Epitope Retrieval Solution 1, Leica, Berlin, Germany) for 20 min by antibody. Single staining with anti-SPP1 (clone HPA027541, 1: 475, Sigma Aldrich, Saint Louis, USA) antibody was performed with a pH9 buffer solution (EDTA buffer, Bond Epitope Retrieval Solution 2, Leica, Berlin, Germany) for 20 min."

Page 4 Lines 118-119, Results section: *“These intermediate-state cells could be independently validated by immunohistochemistry staining, showing co-expression of ZG marker DAB2 and ZF/ZR marker CYP17A1 (Fig. 2i).”*

- Validation of tumor immune microenvironment signatures using spatial transcriptomics and IHC staining:

New Fig. 7d: High definition spatial transcriptomics of ACC18 [...]

New Fig. 7e: Immunohistochemistry staining in two ACC [...]

New Extended Data Fig. 8: High definition spatial transcriptomics of adrenocortical carcinoma.

Page 16 Lines 542-543, Methods section: *“High definition spatial transcriptomics was performed for one ACC sample following the Visium HD Spatial Gene Expression protocol (10x Genomics).”*

Page 17 Line 571-578, Methods section: *“Immunohistochemistry analyses [...].”*

Page 6 Line 189-191, Results section: *“TAM markers were validated in situ using high definition spatial transcriptomics and immunohistochemistry staining (Fig. 7d-e, Extended Data Fig. 8a-b).”*

2. Figure 3c looks exactly the same as Figure 1a, so Figure 3c shows tumor cells or total cells? I think if the author wants to concentrate only on tumor cells in this section, tumor cells should be extracted, re-classified and re-annotated to avoid interference from other normal cells.

Following this reviewer’s comment, Figure 3 now focuses exclusively on tumor steroid cells, after excluding normal steroid cells and non-steroid cells:

Modified Fig. 3c: UMAP of the tumor steroid cells [...]

Modified Fig. 3d: Proportion of main gene modules assigned to steroid cells of each tumor sample.

Page 5 Line 131, Results section: *“Each of the 8 gene modules was scored in each cell of each adrenal tumor sample”*

3. In figure 2i, the number of cells in the intermediate state appears to be significantly less than that in the figure 2a. Please explain why spatial transcriptome sequencing captures fewer intermediate cells than single-nucleus sequencing. Is it a technical problem or something else?

We thank the reviewer for this comment, raising the difficulty of defining clusters on a probably continuous differentiation process. This point is now emphasized in the manuscript.

Page 4 Lines 115-119, Results section: *“Of note, the proportion of intermediate-state cells in spatial transcriptomics and sn-RNAseq was variable, depending on clustering parameters. This variability may reflect the unclear delineation of intermediate-state cells in a progressive trans-differentiation from ZG to ZF. These intermediate-state cells could be independently validated by immunohistochemistry staining, showing co-expression of ZG marker DAB2 and ZF/ZR marker CYP17A1 (Fig. 2i).”*

4. While the spatial transcriptomics data adds valuable spatial context to the atlas, the manuscript could benefit from deeper integration of this data with the single-nucleus transcriptomic findings. Currently, the spatial analysis appears secondary to the primary single-nucleus data. The authors

should more thoroughly integrate these data with the single-nucleus findings, particularly by showing how spatial heterogeneity may corresponds to functional or clinical differences.

Following this reviewer's comment, we now expended the spatial transcriptomics analysis, by:

- Annotating the morphological heterogeneity of tumors studied by spatial transcriptomics. Different tumor regions showing distinct morphological features were characterized.
- Improving the spatial resolution with high resolution spatial transcriptomics. We now provide one example of a single-cell signature (*SPP1*+ TAM), showing spatial heterogeneity. Co-expression of CD68 confirmed the cell type (macrophages). High resolution spatial transcriptomics also showed association of these macrophages with a specific tissue architecture (fibrosis).

This is reflected in the following sections of the manuscript:

New Fig. 7d: High definition spatial transcriptomics of ACC18 [...]

New Extended Data Fig. 8: High definition spatial transcriptomics of adrenocortical carcinoma.

a) Hematoxylin eosin coloration of the ACC18 sample used for high definition spatial transcriptomics.

b) Spatial representation of steroid (*CYP17A1*), fibroblasts (*FN1*) and TAM (*CD68*, *SPP1*) markers.

Modified Extended Data Fig. 9c. Hematoxylin eosin coloration of the ACC2 sample used for spatial transcriptomics. Cyto-nuclear atypia are more pronounced in tumor region 1.

Page 6 Lines 189-191, Results section: "*TAM markers were validated in situ using high definition spatial transcriptomics and immunohistochemistry staining (Fig. 7d-e, Extended Data Fig. 8a-b).*"

Page 8 Lines 228-230, Results section: "*Unsupervised clustering of spatial spots revealed three compartments (cluster1, cluster2, cluster3), corresponding to spatially distinct tumor regions, with different levels of cyto-nuclear atypia (Extended Data Fig. 9c-e).*"

5. The manuscript introduces five distinct ecotypes based on single-nucleus signatures and suggests that these correlate with clinical outcomes. However, it is not clear how these ecotypes were validated externally, particularly across different cohorts. The use of bulk transcriptomic datasets is valuable, but more detail is needed on how well these bulk signatures recapitulate single-nucleus findings.

Following this reviewer's comment, we now provide a validation of single-cell signatures on bulk transcriptome, based on a pseudo-bulk approach. As shown on the plot below, cell proportions estimated by deconvolution and real cell proportions were highly correlated.

This correlation is now reported in the manuscript:

Page 16 Line 509-514, Methods section: "*To evaluate the reliability of deconvolution for resolving each cell subset in bulk tissue, we randomly sampled each population in 1000 cells (if > 2000 cells) or 50% of cells (if < 2000 cells) for training set, used to create the reference matrix for deconvolution, and the remaining cells for testing set, used to create a pseudo-bulk dataset. Cell proportions estimated by deconvolution in snRNA-seq pseudobulk and real cell proportions were highly correlated ($r = 0.71$).*"

In addition, we could validate two single-cell signatures deconvoluted in bulk transcriptomes, using independent biological measurements. For the Eco1 mitoses/hypoxia ecotype, we showed an association with tumor grade (mitotic count or Ki-67). For the Eco2 ZF-like ecotype, we showed an association with cortisol secretion.

This point has been included in the manuscript:

Page 8 Lines 246-248, Results section: “Of note, Eco1 mitosis/hypoxia and Eco2 ZF-like were associated with tumor grade ($p < 10^{-4}$) and cortisol secretion ($p < 10^{-7}$) respectively, supporting the biological relevance of these signatures.”

Additionally, it would strengthen the manuscript to include more direct comparisons to existing ACC prognostic models to clarify the incremental value of the ecotypes.

Following this reviewer’s comment, we now clarified the value of ecotypes compared to existing ACC prognostic models. We evaluated the C-indexes for different prognostic models and compared nested models including clinical variables and ecotypes.

This is now reported in the manuscript:

Page 18 Lines 592-595, Methods section: “To evaluate the value of single-cell signatures compared to existing ACC prognostic models, we computed C-indexes for DFS and OS models based on existing prognostic factors alone and along with single-cell signatures. Comparison of nested models was performed using likelihood-ratio test.”

Page 9 Lines 265-267, Results section: “Finally, compared to models based only on clinical variables, models including ecotypes were better predictors for both DFS (C-index 0.808 vs 0.728, likelihood ratio test LRT $p < 10^{-4}$) and OS (C-index 0.841 vs 0.820, LRT $p < 10^{-3}$, Suppl Table 20).”

Suppl Table 20. C-indexes of disease-free and overall survival models

6. While the identification of ecotypes is novel, the manuscript does not provide enough detail on how the identified ecotypes or gene modules could be applied in clinical practice. The manuscript should clearly outline how these findings could directly influence therapeutic strategies, especially regarding the feasibility of manipulating steroidogenesis and immune cell interactions. Currently, the discussion on therapeutic implications is somewhat speculative and lacks sufficient grounding in experimental data.

We thank this Reviewer for raising this point potentially impacting patients’ care. Though being speculative, this is now deeper discussed in the manuscript.

Page 11 Lines 340-343, Discussion section: “Novel anticortisolic drugs⁵⁸ and combination of these drugs⁵⁹ may be sufficient to reach full blockade of intra-tumor cortisol synthesis. Tumor biopsies after treatment would be needed to ascertain the proper intra-tumor cortisol suppression⁵². If full cortisol blockade could be achieved, would it promote the recruitment of inflammatory macrophages as in Eco5 ZR-like?”

7. The manuscript proposes that modulation of glucocorticoid and androgen levels in tumors could influence immune microenvironments and thus patient outcomes. While intriguing, the proposed mechanism is presented without sufficient experimental validation. The authors should either provide additional evidence supporting this claim or moderate their conclusions to avoid overinterpreting the data.

Following this Reviewer’s comment, we have now moderated the conclusions of the abstract and manuscript.

Page 2 Lines 42-43, Abstract: “These steroid / microenvironment cells interplays may open therapeutic options in aggressive ACC, through immune microenvironment activation by modulating glucocorticoids / androgens balance.”

Page 11 Lines 363-364, Discussion section: “This atlas opens potential perspectives for the treatment of advanced tumors.”

8. The use of single-nucleus sequencing is commendable and appropriate for the study’s objectives. However, the manuscript lacks detail in the description of how quality control was ensured, particularly in terms of potential biases introduced during single-nucleus isolation.

We agree with this Reviewer that biases may arise from the technologies used, whether targeting cells or nuclei isolation, and whether using 10x or another technology. Compared to the recent paper published by Tourigny et al (PMID 38759836), the diversity of microenvironment cells is higher in

our study. In addition, the lymphocyte profile, though depleted, is consistent with priori immunohistochemistry characterization (Landwehr et al, PMID 32474412). Finally, the high proportion of steroid cells (80%) is in agreement with the proportion of tumor cells evaluated by SNP array and exome sequencing studies (Assié et al, PMID 24747642; Zheng et al, PMID 27505681). Thus, we believe that despite potential biases our results are relevant. Following this Reviewer's comment, these potential biases are now better emphasized in the discussion:

Page 10 Lines 303-306, Discussion section: *"While single-nucleus isolation could favor the isolation of certain cell types and bias cell proportions, our results are consistent with previous studies identifying ACC as immune-cold tumors^{10,49}, with lymphocyte depletion. Despite this "cold" immune landscape, this study demonstrates immune variability in ACC, with different levels of exhausted T-cells, pro-inflammatory "M1-like" macrophages and TAM."*

Additionally, more transparency is required in reporting batch effects and how these were controlled, especially for the clustering analysis (UMAP).

We agree with this Reviewer that this point is critical and may considerably impact the findings. For the global cell atlas and for microenvironment analysis, no batch correction was needed, since microenvironment cells from different patients properly aggregated into clusters strongly associated with the normal / benign / malignant nature of the samples. This demonstrates the absence of a major artefactual batch effect. Our strategy to not correct for any batch effect was possible because of minimal unwanted noise, and prevented over-correction that would have suppressed genuine biological differences.

For normal steroid cells, our aim was to define the prototypical signature of each steroid cell type, shared among individuals. Uncorrected UMAP showed the prominent effect of inter-individual differences over steroid cell types differences (Extended Data Fig. 2a-b). We therefore applied batch correction tools, not to correct an artefactual batch effect, but as a way to overcome this inter-individual variability.

For tumor steroid cells, inter-individual variability was also the first determinant of clustering. However, the batch correction tools we tested (Seurat rPCA and cCA, Harmony) all led to over-correction, ascertained by the inappropriate complete mixing of tumor cells with normal cells. Therefore, we could not use batch correction to overcome inter-individual differences. This is why we used an alternative method (e.g. gene modules) to explore the variability of these cell populations.

These points are now detailed in the manuscript:

Page 14 Lines 435-441, Methods section: *"For the global cell atlas and microenvironment analyses, no batch correction was applied, since microenvironment cells properly clustered irrespective of their sample of origin. For normal adrenal steroid cells, data integration was performed to overcome inter-individual variability. We used SCTransform normalization and canonical correlation analysis with default parameters. For tumor steroid cells, data integration tools were tested to overcome inter-individual variability, but led to over-correction. We therefore used another method (recurrent gene modules; see below) to explore intra-tumor variability of tumor steroid cells."*

There is mention of the clustering stability, but the clustering resolution decisions should be more thoroughly explained and justified.

We agree with this Reviewer that clustering resolution should be chosen carefully to avoid over- and underclustering. While some degree of arbitrariness is involved, especially for the identification of cell states - given that these are dynamic and continuous phenomena - our first priority was to select clusters not varying across small clustering resolution changes.

Clustering resolution decisions are now better explained in the manuscript:

Page 13 Lines 427-431, Methods section: “*Clustering resolution was selected on clusters stability using Clustree⁶² v0.5.0.. To achieve the best balance between over- and under-clustering, resolution was chosen based on the following criteria: (i) clusters remain consistent across small clustering resolution changes, (ii) separate groups of cells on UMAP form distinct clusters, and (iii) each cluster shows differentially expressed genes compared to others.*”

Minor comments:

1. In figure 1f, the cells in gray should also be labeled.

Following this Reviewer’s comment, the legend of Fig.1f has been completed.

2. In line 181, some “-” (hyphen) are before the words, some are after the words, and the usage is confusing. It can be used more clearly.

Following this Reviewer’s comment, we have now replaced hyphen by comma:

Page 6 Lines 184-185, Results section: “*Genes associated with pseudotime included NK differentiation markers towards NK-like T cells (KLRD1, KLRF1 and GNLY, Fig. 6f), and THEMIS, a negative regulator of effector CD8⁺ T cells³⁰, towards exhausted T cells (Fig. 6g, Suppl Table 13).*”

3. Figure 8a, two strips (Death event, Cortisol secretion) are in gray and black. It looks a little bit confusing.

Following this Reviewer’s comment, we have now changed the color for Cortisol secretion in Fig. 8a.

4. The study’s identification of immune cell signatures, such as exhausted T cells and inflammatory macrophages, is a critical finding. However, the functional role of these immune cell types in the context of ACC needs further exploration. It is suggested that the authors either include additional functional assays or cite more comprehensive literature to support their conclusions on immune cell behavior and its potential impact on prognosis.

We agree with this Reviewer that the functional characterization of the immune microenvironment remains to be established, beyond the description of ecotypes.

This is now emphasized in the manuscript:

Page 10 Lines 312-319, Discussion section: “*In summary, these ecotypes describe the specific association of steroid cells profiles with immune cell states. The functional importance of tumor immune microenvironment is now well established in response to cytotoxic chemotherapies or immunotherapies in several cancer types^{51,52}. Indeed, cytotoxic chemotherapies induce the release of tumor antigens which recruits and activates antigen-presenting cells, ultimately triggering an antitumor adaptive immune response. And immunotherapies boost the antitumor immune response. To which extent these general oncologic mechanisms apply to ACC remains to be established.*”

.

5. The discussion should more clearly articulate how these findings could be translated into clinical practice. Specifically, the practical steps for using these ecotypes or gene modules as biomarkers for patient stratification should be better outlined.

Following this Reviewer's comment, we now better highlighted the translational relevance of ecotypes.

Page 10 Lines 326-331, Discussion section: *“ACC ecotypes can be leveraged in a translational framework for patient stratification. Cell states provide a valuable prognostic information, applicable to individual patients. A theranostic perspective could also emerge from the association of cell states and response to treatment, which remains to be explored. Determination of cell states statuses could rely on specific markers or on transcriptomic signatures, that can be inferred from bulk transcriptome. This latter approach is increasingly integrated into routine oncology^{54,55} and is applicable to paraffin-embedded samples²².”*

Reviewer #2 (Remarks to the Author): Expert in cancer single-cell and spatial omics, and tumour microenvironment

In this study, Jouinot et al present a single-nucleus transcriptome atlas of 38 human adrenal cortex samples, covering diverse tumor and microenvironment cell types from primary ACCs, ACAs, PBMAHs, and normal adrenals. The dataset is further complemented by spatial transcriptomic profiling of normal and tumor adrenal samples. This atlas enabled the authors to investigate differentiation patterns and functional zonations of normal steroid cells, identify recurrent gene expression programs associated with intratumoral heterogeneity among adrenal tumors, and extract novel gene expression/ecotype signatures correlating with survival.

Overall, the dataset generated in this study provides a substantial resource for the molecular characterization of cellular heterogeneity in diverse adrenal tumors. Existing studies employing similar techniques (snRNA-seq and spatial transcriptomics) have largely focused on single adrenal tumor types or normal adrenal tissues only. The analyses applied (trajectory inference, decomposition of gene expression programs, CNV analysis, etc) are standard and mostly appropriate.

1. Existence of a transitional cell state between ZG and ZF (Line 96). The authors claim to identify an intermediate-state cell cluster that shows features of both ZG and ZF, and that the existence of such an intermediate state supports the gradual differentiation pattern from ZG to ZF cells, as observed in mice lineage tracing studies. Such intermediate-state cells appear to be a substantial proportion (~20-25%) of steroid cells in normal adrenals (Fig. 2a,c). This implies that normal steroid cells continuously undergo lineage conversion en masse under homeostasis, which seems unlikely. For example, this population can potentially be explained by differences in clustering resolutions or cell-type annotation compared to previous studies. Are such intermediate-state cells also observed in previously published adrenal snRNA-seq datasets? The authors were able to map such transitional cell cluster onto the spatial data, but in much fewer numbers (Fig. 2i). How do the authors explain such discrepancy in cell proportions? Also, since the Visium data is not single-cell resolution, such intermediate state spots can be explained by mixing of ZG and ZF cells. Overall, there is limited evidence that such a transitional population exists. It would be convincing to validate the existence and numbers of such intermediate-state cells via immunofluorescent staining.

We thank this Reviewer for questioning the existence of intermediate-state cells, an important question also raised by Reviewer 1. We now provide direct evidence of these cells co-expressing ZG and ZF markers, using immunohistochemistry staining.

The discrepancy in intermediate-state cells proportions between SnRNA-seq and spatial transcriptomics, also raised by Reviewer 1, may reflect the difficulty of defining clusters on a probably continuous differentiation process.

Regarding previously published adrenal snRNA-seq datasets, the work by Iwahashi et al. (PMID 35796577) also reports a cluster of cells co-expressing ZG markers together with ZF/ZR marker CYP17A1, referred to as “ZG-to-ZF transitional cells”, in human adrenal cortex.

These points have been added in the manuscript.

New Fig. 2i: DAB2 (ZG marker, brown) and CYP17A1 (ZF/ZR marker, red) immunohistochemistry staining [...]

Page 17 Lines 571-578, Methods section: “*Immunohistochemistry analyses*

Immunohistochemistries were performed on dewaxed 3 μm slides using a leica BOND III device (Leica, Berlin, Germany). Double staining with anti-DAB2 (clone HPA028888, 1:800, Sigma Aldrich, Saint Louis, USA) and anti-CYP17A1 (clone HPA048533, 1:800, Sigma Aldrich, Saint Louis, USA) antibodies was carried out with a pH6 buffer solution (EDTA buffer, Bond Epitope Retrieval Solution 1, Leica, Berlin, Germany) for 20 min by antibody. Single staining with anti-SPP1 (clone HPA027541, 1: 475, Sigma Aldrich, Saint Louis, USA) antibody was performed with a pH9 buffer solution (EDTA buffer, Bond Epitope Retrieval Solution 2, Leica, Berlin, Germany) for 20 min.”

Page 4 Lines 115-119, Results section: “*Of note, the proportion of intermediate-state cells in spatial transcriptomics and sn-RNAseq was variable, depending on clustering parameters. This variability may reflect the unclear delineation of intermediate-state cells in a progressive trans-differentiation from ZG to ZF. These intermediate-state cells could be independently validated by immunohistochemistry staining, showing co-expression of ZG marker DAB2 and ZF/ZR marker CYP17A1 (Fig. 2i).*”

2. Differentiation states of tumor cells that resemble adrenocortical zonations. Via analysis of gene modules extracted using PCA, the authors find that functional zonation of the normal adrenal cortex is also recapitulated in tumor cells (Fig. 3). Is there evidence of lineage conversion between different tumor cell states (ZG/ZF/ZR-like tumor cells)? How dynamic is this process? This can be addressed by looking at the distribution of tumor cell states along subclonal phylogeny reconstructed by CNVs from snRNA-seq.

We agree with this Reviewer that the co-existence of different steroid cells signatures within a single tumor raises the question of the underlying mechanisms. A sub-clonal divergence with specific chromosomal alterations is unlikely, given the high purity of ACC. A dynamic process is possible, but through a mechanism diverging from the normal adrenal trans-differentiation.

These important points are now added in the manuscript:

Page 9 Lines 284-296, Discussion section: “*The functional zonation of normal adrenal gland is also retrieved in tumor steroid cells, with GM1_ZG, GM2_ZF1, GM3_ZF2 and GM4_ZR gene modules reflecting zona glomerulosa, zona fasciculata and zona reticularis respectively. The co-existence of these different steroid cells signatures within a single tumor questions a potential cell trans-differentiation in tumors. In normal adrenals, trans-differentiation from ZG towards ZF/ZR is related to transient Wnt-βcatenin pathway activation in ZG, followed by cAMP/PKA pathway activation in ZF/ZR¹. In contrast, in ACC, the Wnt-βcatenin pathway is commonly activated by CTNNB1 mutations, leading to a constitutive activation that cannot be switched off. In addition, the co-existence of different steroid cells signatures in ACC does not seem to be linked to “classical” tumor sub-clonality, as assessed by chromosomal alterations. Indeed, ACC commonly exhibit chromosomal*

alterations with > 90% clonality^{6,7}. This suggests that the observed heterogeneity might instead be due to a functional cellular process leading to different cell states. One hypothesis could be the interplay between the cell cycle and steroidogenesis⁴⁵. Further studies are needed to elucidate the mechanisms of steroid differentiation dynamics in ACC."

3. Spatial transcriptomic analysis of ACC2 (Line 221). It is unclear whether all three clusters represent malignant cells instead of normal/stromal cells. What is the CNV status of spots, which may mark tumor margins? Is there a pathology annotation of tissue regions (aggressive features, tumor margin)? Are certain clusters enriched with aggressive features?

Following this reviewer's comment, we now provide additional evidence supporting the malignant nature of the three clusters. Chromosome alterations in ACC2 ST spots were comparable to that observed in snRNA-seq data, and were not observed in a benign control (NAPACA). Moreover, an expert pathologist reviewed the HE slide used for spatial transcriptomics and provided tissue annotations (see also Reviewer 1 point 1).

This is now included in the manuscript:

New Extended Data Fig 9f: Chromosome alterations inferred using InferCNV in spatial transcriptomics spots of ACC2 and a benign tumor presented as control. Predicted gains are colored in red, predicted losses are colored in blue.

Page 8 Lines 230-231, Results section: *"High level of chromosome alterations confirmed the malignant nature of cells (Extended Data Fig 9f)."*

Page 17 Lines 558-561, Methods section: *"Inferred CNV Analysis from spatial transcriptomics Inference of chromosome alterations was performed with InferCNV¹⁹, using a benign tumor as a reference (also processed with Visium Spatial Gene Expression). Inference was performed in ACC2 and in a benign tumor (control)."*

4. Validation of cell-cell interactions in tumor ecotypes (Line 243). The authors identified a few interesting ligand-receptor interactions between tumor cells and stromal/immune cells. Can these specific cell-cell interactions be validated in the spatial transcriptomics data?

We agree with this reviewer that cell-cell interactions called in snRNA-seq data require validation. Expression levels of ligand and receptor cannot provide proof of interaction. We are not sure that spatial transcriptomics analysis of cell cell interactions will add more evidence, even if spatial proximity is observed. Indeed we would still lack the direct evidence of molecular interaction, which requires cell biology experiments. Although speculative, the identification of potential interactions may provide guidance for future functional studies of these interactions.

We now included these points:

Discussion section, Pages 10-11 Lines 335-337: *"Of note, these interactions are based only on gene expression levels at this stage and are therefore speculative. However, the list of potential interactions may orient future experimental studies in search for novel therapeutic targets."*

5. HSP+ cells. The authors identify a distinct steroid cluster with high expression of heat shock proteins (Fig. 2), potentially reflecting a stress response to ROS production associated with steroidogenesis. This is an interesting observation; is there evidence of DNA damage or chromosomal aberrations in such cells? Can the authors speculate whether these cells could be the cell-of-origin of certain adrenal tumors?

Following this Reviewer’s comment, we have compared the level of chromosomal alterations in HSP+ cells compared to other normal steroid cells and to tumor steroid cells (see figure below). Although we cannot exclude some chromosome alterations, CNV profile was not different from that observed in steroid cells from benign tumors. We have now added this point in the Discussion.

Page 9 Lines 281-283, Discussion section: *“Potentially in line with hypothesis, the inference of chromosome alterations with InferCNV in HSP+ cells showed a slight but limited increase of chromosome alterations (data not shown).”*

Reviewer #3 (Remarks to the Author): Expert in adrenocortical carcinomas

I have reviewed the manuscript by Dr. Jouinot and co-workers, which presents a comprehensive single-cell atlas of the adrenal gland, adrenal tumors and adrenocortical carcinoma. I was impressed by this work that succeeds not only in performing advanced molecular/bioinformatics analyses but also excels in integrating these findings with the current literature in a clear and logical manner. This accomplishment is particularly noteworthy in an era where data is abundant, but refined analyses and well-structured presentations are more scarce. I have only minor comments of how this manuscript may be improved.

1. Abstract: The abstract has a less refined feeling to it than the text in the full article. It is also unclear to the reader what are novel/validatory findings.

Following this Reviewer's comment, we now clarified the novelties in the abstract:

Page 2 Lines 34-35, Abstract: *"We identify intermediate-state cells between glomerulosa and fasciculata, a novel transition state in the centripetal trans-differentiation of normal steroid cells."*

Page 2 Lines 42-44, Abstract: *"These steroid / microenvironment cells interplays improve outcome predictions and may open therapeutic options in aggressive ACC, through immune microenvironment activation by modulating glucocorticoids / androgens balance."*

2. Clinical annotations:

a. Were all tumour samples therapy-naïve?

We thank this Reviewer for this raising this question, as antitumour treatment received before surgery could have biased some analyses. All tumour samples were indeed therapy-naïve, including the metachronous metastasis sample (ACC1b), surgically removed before mitotane initiation.

This information is now included in the Methods:

Page 12 Line 388, Methods section: *"All samples were collected before the initiation of antitumor treatment."*

b. Starting at line 206, the manuscript investigates tumour regions exhibiting "aggressive features." It would be beneficial to clearly define what is meant by "aggressive features".

Following this Reviewer's comment, we more precisely defined « aggressive features ». We also added morphological annotations by an adrenal expert pathologist of all pathology slides reported in the manuscript (see also Reviewer 1 point 1 and Reviewer 2 point 3).

Modified Extended Data Fig. 9c. Hematoxylin eosin coloration of the ACC2 sample used for spatial transcriptomics. Cyto-nuclear atypia are more pronounced in tumor region 1.

Page 8 Lines 228-231, Results section: *"Unsupervised clustering of spatial spots revealed three compartments (cluster1, cluster2, cluster3), corresponding to spatially distinct tumor regions, with different levels of cyto-nuclear atypia (Extended Data Fig. 9c-e)."*

3. Multivariate model: The manuscript reports on the prognostic relevance of "proliferation" and "ZF-like" ecotypes. Im curious to how do these findings align with those already established in clinical practice, specifically regarding tumour proliferation rate (e.g., Ki-67) and hormone secretion? It would be useful to further discuss the connection between these molecular findings and established clinical markers.

Following this Reviewer's comment, we now provide associations between ecotypes and proliferation and secretion. Although Eco1 mitosis/hypoxia and Eco2 ZF-like were associated with tumor grade and cortisol secretion respectively, multivariate models including ecotypes better predict survival than models including only clinical variables (see also Reviewer 1 point 5).

These points have now been added to the manuscript:

Page 8 Lines 246-248, Results section: “Of note, *Eco1* mitosis/hypoxia and *Eco2* ZF-like were associated with tumor grade ($p < 10^{-4}$) and cortisol secretion ($p < 10^{-7}$) respectively, supporting the biological relevance of these signatures.”

Page 18 Lines 592-595, Methods section: “To evaluate the value of single-cell signatures compared to existing ACC prognostic models, we computed C-indexes for DFS and OS models based on existing prognostic factors alone and along with single-cell signatures. Comparison of nested models was performed using likelihood-ratio test.”

Page 9 Lines 265-267, Results section: “Finally, compared to models based only on clinical variables, models including ecotypes were better predictors for both DFS (C-index 0.808 vs 0.728, likelihood ratio test LRT $p < 10^{-4}$) and OS (C-index 0.841 vs 0.820, LRT $p < 10^{-3}$, Suppl Table 20).”

Suppl Table 20. C-indexes of disease-free and overall survival models

4. Figures:

a. Figure 1: The colour scheme used to represent different cell types can sometimes be challenging to follow, particularly when distinguishing between Chromaffin and Myeloid cells. In panel C, the order in which the cell types appear differs from panels A and B, which may contribute to confusion. It would be helpful to harmonize the presentation across all panels.

Colors of Fig. 1a, 1b and 1c have been modified according to the Reviewer’s comment.

b. Figure 2: The legend could be improved to better guide the reader in interpreting the data. For example, I assume that a "dot" represents a “cell” in panels A, D, and F, and that the black line in panel F indicates the mean or median. Explicitly stating these points in the figure legend would enhance the clarity for readers not experts in interpreting data from single cell experiments.

Figures legends have been modified according to the Reviewer’s comment.

Modified Fig. 1: Cell atlas of normal adrenal cortex and adrenocortical tumors.

Individual cells from 34 adrenocortical tumors and 4 normal adrenals are presented (single-nucleus transcriptomes).

Modified Fig. 2: Characterization of normal steroid cells.

Individual steroid cells from 4 normal adrenals are presented (single-nucleus transcriptomes)

Modified Fig. 2f: Expression profiles of selected genes with pseudotime variation, including genes related to calcium signaling (*ATP10A*, *CACNB2*, *CALNI*), to Wnt- β catenin signaling (*AFF3*, *LEF1*, *LGR5*), to cell adhesion (*ITGAI*, *NCAMI*, *VCAN*), and to steroidogenesis (*CYP11B2*, *SCARB1*, *CYP11B1*, *STAR*, *CYB5A*, *SULT2A1*). Black lines: mean expression over pseudotime.

Modified Figure 3c: UMAP of the steroid tumor cells. Individual steroid cells from 34 adrenocortical tumors are presented (single-nucleus transcriptomes)

Modified Fig. 4: Characterization of adrenocortical fibroblasts.

Individual fibroblasts cells from 34 adrenocortical tumors and 4 normal adrenals are presented (single-nucleus transcriptomes).

Modified Fig. 4g: Expression profiles of top genes with pseudotime variation from resident fibroblasts towards CAF3 (solid line), including pro-angiogenic (*RGS5*, *SEMA5A*) and immunosuppressive markers (*CD36*). Black lines: mean expression over pseudotime.

Modified Fig. 5: Characterization of adrenocortical endothelial cells.

Individual endothelial cells from 34 adrenocortical tumors and 4 normal adrenals are presented (single-nucleus transcriptomes).

Modified Fig. 5f: Expression profiles of top genes with pseudotime variation from EC-venous towards TEC2 (dashed line), including known TEC-associated genes (*VWF*, *ANGPT2*) and novel markers (*ANO2*, *KCNQ3*, *LAMB1*, *ENPP2*). Black lines: mean expression over pseudotime.

Modified Fig. 6: Characterization of adrenocortical lymphocytes.

Individual lymphoid cells from 34 adrenocortical tumors and 4 normal adrenals are presented (single-nucleus transcriptomes).

Modified Fig. 6f: Expression profiles of top genes with pseudotime variation, from unassigned T cells towards NK-like T cells (solid line), including NK differentiation markers (*KLRD1*, *KLRF1* and *GZMB*). Black lines: mean expression over pseudotime.

Modified Fig. 6g: Expression profiles of top genes with pseudotime variation, from unassigned T cells towards exhausted T cells (dashed line), including *THEMIS*, a negative regulator of TCR signaling. Black lines: mean expression over pseudotime.

Modified Fig. 7: Characterization of adrenocortical myeloid cells.

Individual myeloid cells from 34 adrenocortical tumors and 4 normal adrenals are presented (single-nucleus transcriptomes).

Modified Fig. 7h: Expression profiles of top genes with pseudotime variation from resident macrophages towards TAM1 (solid line), including M2/TAM markers (*PPARG*, *GPNMB*, *CCL18*), and anti-inflammatory markers including cholesterol transporters (*ABCA1*, *ABCG1*) and the metalloproteinase *MMP19*. Black lines: mean expression over pseudotime.

Modified Fig. 7i: Expression profiles of top genes with pseudotime variation from resident macrophages towards inflammatory macrophages (dashed line), including pro-inflammatory markers (*CX3CR1*, *C3*, *PRKG1*) and the adhesion G protein-coupled receptor *ADGRB3*. Black lines: mean expression over pseudotime.